# PPGFlowECG: Latent Rectified Flow with Cross-Modal Encoding for PPG-Guided ECG Generation and Cardiovascular Disease Detection

## Abstract

In clinical practice, electrocardiography (ECG) remains the gold standard for cardiac monitoring, providing crucial insights for diagnosing a wide range of cardiovascular diseases (CVDs). However, its reliance on specialized equipment and trained personnel limits feasibility for continuous routine monitoring. Photoplethysmography (PPG) offers accessible, continuous monitoring but lacks definitive electrophysiological information, preventing conclusive diagnosis. Generative models present a promising approach to translate PPG into clinically valuable ECG signals, yet current methods face substantial challenges, including the misalignment of physiological semantics in generative models and the complexity of modeling in high-dimensional signals. To this end, we propose PPGFlowECG, a two-stage framework that aligns PPG and ECG in a shared latent space via the CardioAlign Encoder and employs latent rectified flow to generate ECGs with high fidelity and interpretability. To the best of our knowledge, this is the first study to experiment on MCMED, a newly released clinical-grade dataset comprising over 10 million paired PPG–ECG samples from more than 118,000 emergency department visits with expert-labeled cardiovascular disease annotations. Results demonstrate the effectiveness of our method for PPG-to-ECG translation and cardiovascular disease detection. Moreover, cardiologist-led evaluations confirm that the synthesized ECGs achieve high fidelity and improve diagnostic reliability, underscoring our method's potential for real-world cardiovascular screening. The project is publicly available at `https://anonymous.4open.science/r/PPGFlowECG-D6F3`.

## 1 Introduction

Cardiovascular diseases (CVDs) are the leading cause of death worldwide, claiming millions of lives annually (Timmis et al., 2022; Vaduganathan et al., 2022; Chong et al., 2024). Continuous cardiac monitoring offers a promising means for early detection and timely intervention (Bayoumy et al., 2021). Electrocardiography (ECG), the clinical gold standard, provides detailed electrophysiological information but relies on specialized equipment and trained personnel, limiting routine use (Kligfield et al., 2007). Although recent advances have yielded wearable alternatives (Xie et al., 2024; Li et al., 2025), key limitations persist. For instance, the Apple Watch records only 30 seconds of data and requires active hand contact, restricting its value for continuous monitoring (Nelson et al., 2020). To overcome these limitations, photoplethysmography (PPG) has emerged as a non-invasive, cost-effective modality for continuous cardiac monitoring. As an optical technique that measures blood volume

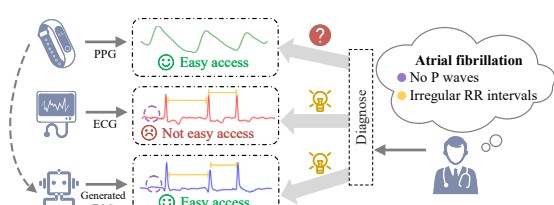

Figure 1: PPG is easy to acquire but lacks diagnostic fidelity, whereas ECG reveals definitive disease markers. AI-based PPG-to-ECG translation offers a promising diagnostic pathway.

changes to infer cardiac activity, PPG offers superior accessibility and comfort compared to ECG, enabling long-term use without skin irritation. Recent advances have demonstrated its value in applications such as motion-robust heart rate estimation, cuffless blood pressure assessment, and short-term heart rate variability prediction(Zhang et al., 2014; Kim et al., 2022; Gupta et al., 2025). However, unlike ECG, PPG lacks detailed electrophysiological information, restricting its role to physiological monitoring rather than definitive disease diagnosis (Pereira et al., 2020). As shown in Figure 1, this limitation underscores the promise of PPG-to-ECG translation, which seeks to recover clinically relevant ECG features from PPG signals via cross-modal mapping, thereby enhancing cardiovascular disease screening and diagnosis.

The emergence of large-scale public datasets such as MCMED(Kansal et al., 2025), which provide synchronized PPG–ECG recordings with reliable CVD annotations, has created transformative opportunities for advancing PPG-to-ECG translation and improving cardiovascular disease diagnosis. A promising technical direction is the application of diffusion models(Ho et al., 2020) and flow models(Liu et al., 2022). While diffusion models exhibit strong generative capacity, they require hundreds of iterative denoising steps, resulting in slow inference. Flow models address this inefficiency by learning a direct transformation from noise to target along a single trajectory.

Despite recent progress, PPG-to-ECG translation faces two major challenges: 1) *Physiological Semantics Misalignment in Generative Models*: Conventional end-to-end approaches often act as mere "waveform mimickers," focusing on low-level signal reconstruction rather than physiological semantics. As a result, generated ECGs may resemble authentic waveforms morphologically but fail to preserve clinically actionable features essential for reliable cardiovascular disease screening and diagnosis, thereby limiting practical utility. 2) *Modeling Complexity in High-Dimensional Signals*: Complex temporal dependencies, inter-subject variability, and abrupt waveform transitions create highly non-smooth, modality-specific signal manifolds in raw data space. These properties violate the continuity and invertibility assumptions underlying conventional generative models, particularly flow-based architectures, leading to unstable training and poor generalization.

To address these challenges, we propose PPGFlowECG, a two-stage framework that combines cross-modal encoding with latent rectified flow for PPG-to-ECG translation and cardiovascular disease detection. In Stage 1, we pretrain a cross-modal encoder–decoder built around a shared CardioAlign Encoder, which processes both PPG and ECG with identical parameters. This design compels the model to move beyond superficial waveform similarity and capture modality-invariant cardiovascular dynamics. In Stage 2, a rectified flow model is trained within the latent space to learn the shortest deterministic trajectory from Gaussian noise to the target ECG latent representation. This "Align first, Generate later" paradigm enables ECG generation with high morphological fidelity and semantic equivalence to real signals, thereby ensuring clinical interpretability and diagnostic utility. The main contributions of this work are summarized in below:

- We propose PPGFlowECG, a novel two-stage framework for high-fidelity PPG-to-ECG translation and cardiovascular disease detection. To our knowledge, this is the first latent rectified flow model specifically designed for cross-modal physiological signal generation.

- We introduce an "Align First, Generate Later" paradigm, in which a shared CardioAlign Encoder learns a semantically aligned latent space across PPG and ECG. Within this space, a latent rectified flow model deterministically maps Gaussian noise to ECG representations along the shortest trajectory.

- We conduct the first large-scale evaluation on MCMED, a clinical-grade dataset comprising over 10 million paired PPG–ECG samples from more than 118,000 emergency department visits with expert-labeled CVD annotations. Results demonstrate the effectiveness of our approach in both signal translation and disease detection.

- Finally, cardiologist-led clinical evaluations confirm that the synthesized ECGs exhibit high fidelity and enhance diagnostic reliability, underscoring our framework's potential for real-world cardiovascular screening.

## 2 RELATED WORK

Recent advances have fueled growing interest in modeling ECG signals from PPG data using generative approaches. Early work relied on traditional machine learning techniques to infer key ECG

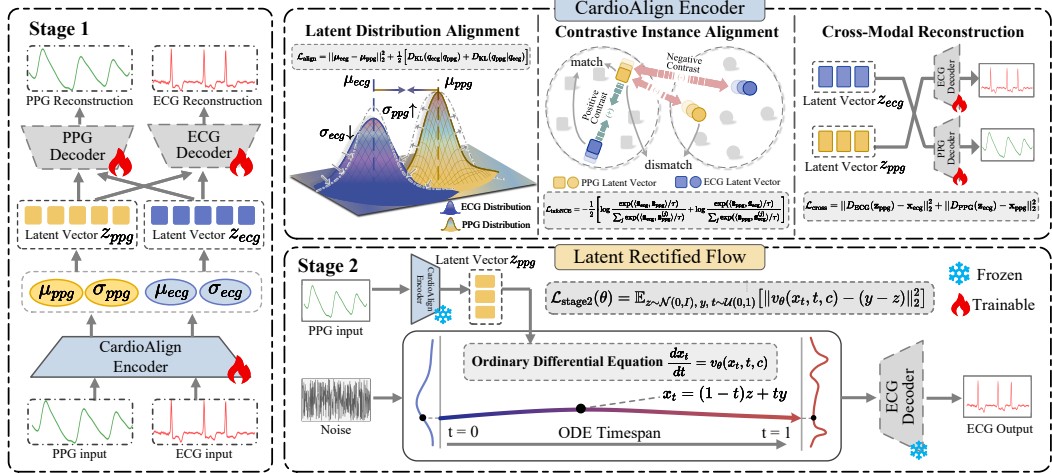

Figure 2: Illustration of the proposed PPGFlowECG framework for high-fidelity PPG-to-ECG translation and cardiovascular disease detection. In this figure, the framework aligns PPG and ECG in a shared latent space using the CardioAlign Encoder and employs latent rectified flow to synthesize ECGs with high fidelity and interpretability.

features from PPG waveforms (Banerjee et al., 2014; Tian et al., 2022; Zhu et al., 2021). For example, Zhu et al. (2021) applied the Discrete Cosine Transform (DCT) to approximate mappings between the two modalities. However, such handcrafted pipelines fail to capture complex nonlinear dependencies and are prone to bias when scaled to raw, heterogeneous signals. To overcome these limitations, research has shifted toward end-to-end deep generative models. CardioGAN (Sarkar & Etemad, 2021), for instance, employs a CycleGAN-based architecture for PPG-to-ECG translation under both paired and unpaired supervision, drawing inspiration from image-to-image translation. P2E-WGAN (Vo et al., 2021) adopts a conditional GAN with a Wasserstein loss to improve stability and reconstruction quality. ADSSM (Vo et al., 2024) employs a variational inference framework with a Gaussian prior to approximate the posterior distribution, while the Region-Disentangled Diffusion Model (RDDM) (Shome et al., 2024) leverages diffusion processes to capture spatially localized waveform features, enhancing morphological realism. Beyond translation, specialized models have been developed for downstream cardiovascular disease detection. For example, Performer (Lan, 2023) reconstructs ECGs with transformer architectures and integrates them with native PPG to improve diagnostic performance.

## 3 METHODOLOGY

### 3.1 OVERVIEW OF PPGFLOWECG

PPGFlowECG generates ECG from PPG in two stages: (i) a shared CardioAlign Encoder establishes a semantically aligned PPG–ECG latent space, and (ii) a latent rectified flow model conditioned on PPG deterministically transports Gaussian noise along the shortest semantic trajectory to an ECG latent, which the decoder converts into a waveform. As illustrated in Figure 2, this framework mitigates the instability induced by non-smooth, modality-specific structures in raw signal space while preserving morphological fidelity and clinical interpretability in the synthesized ECGs. The training and sampling algorithms are provided in Appendix B.

### 3.2 CARDIOALIGN ENCODER

PPG and ECG exhibit distinct surface morphologies yet provide complementary perspectives on the same underlying cardiovascular dynamics. When trained independently, modality-specific encoders often exploit superficial shortcuts and neglect shared physiological semantics. The central innovation of the CardioAlign Encoder is to regard PPG and ECG not as separate languages but as dialects of a common physiological interlingua. By enforcing parameter sharing, the encoder $E_{CA}(\cdot)$ aligns

both modalities in a unified latent space, transcending superficial waveform differences to capture modality-invariant cardiovascular dynamics and preserve clinically meaningful features essential for downstream diagnosis. As illustrated in Figure 2, the stage 1 adopts an encoder–decoder architecture centered on the CardioAlign Encoder. A single encoder $E_{\mathrm{CA}}(\cdot)$ processes both PPG and ECG inputs, while modality-specific decoders $D_{\mathrm{PPG}}(\cdot)$ and $D_{\mathrm{ECG}}(\cdot)$ reconstruct their respective signals. For an input waveform $x_m \in \mathbb{R}^{L \times 1}$ of length $L$ from modality $m \in \{\mathrm{ppg}, \mathrm{ecg}\}$, the encoder produces Gaussian parameters $\boldsymbol{\mu}_m$ and $\boldsymbol{\sigma}_m$, from which the latent representation $\mathbf{z}_m$ is sampled via the reparameterization trick:

$$\mathbf{z}_m = \boldsymbol{\mu}_m + \boldsymbol{\sigma}_m \odot \boldsymbol{\epsilon}, \quad \boldsymbol{\epsilon} \sim \mathcal{N}(\mathbf{0}, \mathbf{I}). \tag{1}$$

To enforce modality-invariant cardiovascular dynamics in the latent space, the CardioAlign Encoder is trained with a set of complementary alignment objectives.

**Latent Distribution Alignment.** We initiate alignment at the distributional level by modeling each modality's latent posterior as a Gaussian. Specifically, the posteriors for PPG and ECG, denoted as $q_{\mathrm{ppg}}$ and $q_{\mathrm{ecg}}$, are defined as:

$$q_{\mathrm{ppg}} = \mathcal{N}(\boldsymbol{\mu}_{\mathrm{ppg}}, \mathrm{diag}(\boldsymbol{\sigma}_{\mathrm{ppg}}^2)), \quad q_{\mathrm{ecg}} = \mathcal{N}(\boldsymbol{\mu}_{\mathrm{ecg}}, \mathrm{diag}(\boldsymbol{\sigma}_{\mathrm{ecg}}^2)), \tag{2}$$

where $\boldsymbol{\mu}_{\mathrm{ppg}}$ and $\boldsymbol{\mu}_{\mathrm{ecg}}$ are the posterior means, and $\boldsymbol{\sigma}_{\mathrm{ppg}}^2, \boldsymbol{\sigma}_{\mathrm{ecg}}^2$ are the corresponding variances for PPG and ECG, respectively. To align these distributions, $\mathcal{L}_{\mathrm{align}}$ is defined as follow:

$$\mathcal{L}_{\mathrm{align}} = ||\boldsymbol{\mu}_{\mathrm{ecg}} - \boldsymbol{\mu}_{\mathrm{ppg}}||_2^2 + \tfrac{1}{2}\Big[D_{\mathrm{KL}}(q_{\mathrm{ecg}}|q_{\mathrm{ppg}}) + D_{\mathrm{KL}}(q_{\mathrm{ppg}}|q_{\mathrm{ecg}})\Big], \tag{3}$$

where the first term enforces mean consistency by encouraging the centroids of the two modalities to converge, while the symmetric Kullback–Leibler (KL) divergence (Zhang et al., 2023) aligns their variance structures. Together, these constraints provide coarse-grained yet stable alignment, preventing distributional collapse and fostering a shared physiological latent space.

**Contrastive Instance Alignment.** While latent distribution alignment provides coarse-grained consistency, we further refine alignment at the instance level using an InfoNCE loss (He et al., 2020) applied to mean-pooled latent representations. Let $\bar{\mathbf{z}}_{\mathrm{ecg}}$ and $\bar{\mathbf{z}}_{\mathrm{ppg}}$ denote the pooled representations of ECG and PPG, respectively. Temporally paired samples are treated as positives, whereas all other combinations serve as negatives,

$$\mathcal{L}_{\mathrm{infoNCE}} = -\frac{1}{2}\left[\log \frac{\exp(\langle \bar{\mathbf{z}}_{\mathrm{ecg}}, \bar{\mathbf{z}}_{\mathrm{ppg}}\rangle/\tau)}{\sum_j \exp(\langle \bar{\mathbf{z}}_{\mathrm{ecg}}, \bar{\mathbf{z}}_{\mathrm{ppg}}^{(j)}\rangle/\tau)} + \log \frac{\exp(\langle \bar{\mathbf{z}}_{\mathrm{ppg}}, \bar{\mathbf{z}}_{\mathrm{ecg}}\rangle/\tau)}{\sum_j \exp(\langle \bar{\mathbf{z}}_{\mathrm{ppg}}, \bar{\mathbf{z}}_{\mathrm{ecg}}^{(j)}\rangle/\tau)}\right]. \tag{4}$$

Here, $\langle \cdot, \cdot \rangle$ denotes dot-product similarity and $\tau$ is a temperature parameter. This contrastive regularization sharpens pairwise alignment by pulling matched ECG–PPG pairs closer and pushing mismatched pairs apart, thereby ensuring that the latent space remains instance-discriminative while preserving subject-specific cardiovascular semantics.

**Cross-Modal Reconstruction.** While the above objectives establish a coarse-to-fine alignment between modalities, a truly unified representation should not only inhabit a shared latent space but also operate within a common language. To impose this stronger functional constraint, we require the latent representation of one modality to faithfully reconstruct the other. This principle of cross-modal decodability ensures that the latent space encodes core, translatable physiological information essential for genuine modality fusion. $\mathcal{L}_{\mathrm{cross}}$ is defined as follow:

$$\mathcal{L}_{\mathrm{cross}} = ||D_{\mathrm{ECG}}(\mathbf{z}_{\mathrm{ppg}}) - x_{\mathrm{ecg}}||_2^2 + ||D_{\mathrm{PPG}}(\mathbf{z}_{\mathrm{ecg}}) - x_{\mathrm{ppg}}||_2^2. \tag{5}$$

Here, $x_{\mathrm{ecg}}$ and $x_{\mathrm{ppg}}$ denote the ground-truth signals. By enforcing $\mathbf{z}_{\mathrm{ppg}}$ to decode into ECG and $\mathbf{z}_{\mathrm{ecg}}$ into PPG, the latent space is driven to capture physiologically meaningful factors that generalize across modalities, thereby reinforcing cross-modal translatability.

**Training and Optimization.** In standard variational autoencoders (Kingma & Welling, 2013), the loss consists of a reconstruction term $\mathcal{L}_{\mathrm{rec}}$ and a KL regularizer $\mathcal{L}_{\mathrm{kl}}$. Building on this baseline, we introduce a unified CardioAlign Loss ($\mathcal{L}_{\mathrm{CA}}$), which integrates latent distribution alignment, contrastive instance alignment, and cross-modal reconstruction. The overall training objective is formulated as:

$$\mathcal{L}_{\mathrm{stage1}} = \sum_{m \in \mathcal{M}} \Big[\mathcal{L}_{\mathrm{rec}}^{(m)} + \alpha\, \mathcal{L}_{\mathrm{kl}}^{(m)}\Big] + \mathcal{L}_{\mathrm{CardioAlign}}, \quad \mathcal{M} = \{\mathrm{ppg}, \mathrm{ecg}\}, \tag{6}$$

where $\alpha$ is a weighting coefficient for the KL term, typically set to $10^{-4}$.

## 3.3 LATENT RECTIFIED FLOW

To achieve high-fidelity ECG generation from PPG, we develop a generative model based on latent rectified flow. The key advantage of this approach is that it reduces the generative process to learning a straight-line vector field in a well-structured latent space. Instead of operating in the raw signal domain, we leverage the latent space aligned by the CardioAlign Encoder, which is lower-dimensional and smoother, thereby better satisfying the continuity assumptions of flow models and markedly improving both training stability and sampling efficiency.

**Generative Modeling of Latent Representations.** Our objective is to learn a conditional vector field $v_\theta$ that transports an isotropic Gaussian prior $\mathcal{N}(\mathbf{0}, \mathbf{I})$ to the target ECG latent distribution. Let $y$ denote the ECG latent representation produced by the CardioAlign Encoder, and $c$ the corresponding PPG latent representation used as the conditioning signal. Starting from a noise sample $z \sim \mathcal{N}(\mathbf{0}, \mathbf{I})$, the model learns a deterministic mapping from $z$ to $y$ guided by $c$. The central insight of the Rectified Flow framework is that the generative process can be reduced to learning a straight-line trajectory from $z$ to $y$. For any interpolation time $t \sim \mathcal{U}(0, 1)$, the intermediate latent state is defined as:

$$x_t = (1 - t)z + ty. \tag{7}$$

Here, $t = 0$ corresponds to the initial noise and $t = 1$ to the target. Because the trajectory is linear, the ideal drift—i.e., the instantaneous velocity vector—remains constant and independent of $t$:

$$v^\star(x_t, t, c) = y - z, \tag{8}$$

where $v^\star$ denotes the ground-truth field defined by the linear trajectory, and the learnable field $v_\theta$ is trained to approximate it. This formulation removes the need to capture complex time-dependent dynamics, thereby simplifying the learning process. To estimate this constant drift, the parameterized model $v_\theta(x_t, t, c)$ is optimized with a mean-squared error (MSE) objective:

$$\mathcal{L}_{\text{stage2}}(\theta) = \mathbb{E}_{z \sim \mathcal{N}(0, I),\, y,\, t \sim \mathcal{U}(0, 1)} \big[ \| v_\theta(x_t, t, c) - (y - z) \|_2^2 \big]. \tag{9}$$

In practice, stability is improved by sampling multiple independent time points $t$ for each $(z, y)$ pair. This strategy retains the simplicity of rectified flow while promoting robust convergence, making it particularly effective for generative modeling in structured latent spaces.

**Conditioning Mechanisms.** To accurately learn the conditional vector field $v_\theta$, we employ a Transformer architecture explicitly guided by the PPG latent representation. The backbone begins with a one-dimensional convolutional token embedder that projects the ECG latent $y$ into a sequence of compact tokens while preserving local temporal structure. These tokens are enriched with learnable positional encodings to provide temporal ordering and are subsequently processed by a Transformer encoder that captures long-range physiological dependencies. Conditioning is introduced via a cross-attention mechanism in every Transformer block, where the PPG latent $c$ serves as the key–value pair and the evolving ECG representation serves as the query. This design enables dynamic alignment of relevant PPG regions with the evolving ECG trajectory, thereby guiding the prediction of the conditional vector field $v_\theta(x_t, t, c)$. Incorporating $c$ at every layer, rather than only at the input, ensures persistent cross-modal interaction throughout the network. This hierarchical conditioning strategy facilitates fine-grained temporal alignment between modalities and enhances the model's ability to learn physiologically consistent transformations from PPG to ECG.

**Inference and Sampling.** During inference, the objective is to synthesize a new ECG signal conditioned on the PPG latent $c$. We begin by sampling an initial noise vector $z \sim \mathcal{N}(\mathbf{0}, \mathbf{I})$, which serves as the starting point of the generative trajectory. The transformation from $z$ to the target ECG latent $y$ is governed by an Ordinary Differential Equation (ODE) (Hartman, 2002) defined by the learned conditional vector field $v_\theta$:

$$\frac{dx_t}{dt} = v_\theta(x_t, t, c), \quad t \in [0, 1], \tag{10}$$

where $x_t$ denotes the latent state at time $t$ and $c$ is the conditioning PPG latent. To approximate this continuous trajectory, we adopt an explicit Euler solver with a fixed number of steps $T$. Let $\Delta t = \frac{1}{T}$ and $t_k = k \cdot \Delta t$. The latent state is iteratively updated as:

$$x_{k+1} = x_k + \Delta t \cdot v_\theta(x_k, t_k, c), \qquad k = 0, 1, \ldots, T - 1, \quad x_0 \sim \mathcal{N}(\mathbf{0}, \mathbf{I}). \tag{11}$$

After integrating the ODE from $t = 0$ to $t = 1$, the terminal state $x_T$ is obtained as the generated ECG latent representation, which is then decoded by the frozen Stage 1 ECG decoder to reconstruct the synthetic waveform in the signal domain.

Table 1: Performance comparison on different datasets. For all metrics, lower values indicate better performance. The best result for each metric is **bolded**, and the second-best is underlined. T denotes the number of sampling steps.

| Datasets | Methods | MAE | RMSE | FD | FID | MAE$_{HR}$ |
|---|---|---|---|---|---|---|
| MCMED | CardioGAN (Sarkar & Etemad, 2021) | 0.98 | 1.40 | 80.19 | 53.54 | 2.72 |
| | DDPM (T = 50) (Ho et al., 2020) | 0.94 | 1.36 | **13.02** | 44.72 | 7.00 |
| | RDDM (T = 10) (Shome et al., 2024) | 0.99 | 1.41 | 56.19 | 20.81 | 7.45 |
| | Rectified Flow (T = 10) (Liu et al., 2022) | 0.97 | 1.36 | 104.35 | 54.71 | 26.40 |
| | **PPGFlowECG (T = 10)** | **0.73** | **1.14** | 43.99 | **12.84** | **1.80** |
| VitalDB | CardioGAN (Sarkar & Etemad, 2021) | 0.82 | 1.32 | 88.97 | **20.92** | 3.50 |
| | DDPM (T = 50) (Ho et al., 2020) | 0.83 | 1.31 | **40.76** | 32.03 | 11.57 |
| | RDDM (T = 10) (Shome et al., 2024) | 0.87 | 1.40 | 131.74 | 64.62 | 30.27 |
| | Rectified Flow (T = 10) (Liu et al., 2022) | 0.90 | 1.37 | 90.59 | 64.23 | 36.36 |
| | **PPGFlowECG (T = 10)** | **0.59** | **1.10** | 54.87 | 27.09 | **3.23** |
| MIMIC-AFib | CardioGAN (Sarkar & Etemad, 2021) | **0.73** | **1.22** | 64.19 | 45.86 | 3.42 |
| | DDPM (T = 50) (Ho et al., 2020) | **0.73** | **1.22** | 107.20 | 42.86 | 9.55 |
| | RDDM (T = 10) (Shome et al., 2024) | 0.82 | 1.37 | 77.77 | 42.57 | 15.35 |
| | Rectified Flow (T = 10) (Liu et al., 2022) | 0.83 | 1.36 | 68.28 | 49.30 | 26.01 |
| | **PPGFlowECG (T = 10)** | **0.73** | 1.29 | 63.75 | **37.69** | **2.30** |
| BIDMC | CardioGAN (Sarkar & Etemad, 2021) | 0.82 | 1.31 | 79.64 | 63.35 | **1.38** |
| | DDPM (T = 50) (Ho et al., 2020) | 0.79 | 1.24 | 135.88 | 54.88 | 5.82 |
| | RDDM (T = 10) (Shome et al., 2024) | 0.83 | 1.34 | 89.81 | 63.06 | 10.23 |
| | Rectified Flow (T = 10) (Liu et al., 2022) | 0.82 | 1.33 | 74.48 | 55.54 | 9.14 |
| | **PPGFlowECG (T = 10)** | **0.71** | **1.15** | **46.72** | 54.22 | 2.35 |

## 4 EXPERIMENTS

### 4.1 EXPERIMENT SETUP

**Datasets.** We conduct experiments on four representative datasets encompassing diverse clinical scenarios: MCMED (Kansal et al., 2025), VitalDB (Lee et al., 2022), MIMIC-AFib (Bashar et al., 2019), and BIDMC (Pimentel et al., 2016). MCMED is the largest clinical-grade dataset, comprising over 10 million paired PPG–ECG samples from more than 118,000 emergency department visits with expert-labeled cardiovascular disease annotations. For consistency and clinical relevance in rhythm analysis, Lead II ECG is uniformly adopted as the reference channel across all datasets.

**Data Processing.** For ECG and PPG preprocessing, standardized procedures were applied to ensure consistency and data quality (Pan & Tompkins, 2007). Continuous signals were segmented into non-overlapping 10-second windows, with segments containing missing values or excessive flatness discarded. ECG signals were high-pass filtered at 0.5 Hz, while PPG signals underwent band-pass filtering between 0.5 and 8 Hz. Signal quality was assessed separately: ECG windows using SQI methods from NeuroKit2 (Makowski et al., 2021), and PPG windows using peak detection and template matching. Low-quality segments were excluded. To harmonize datasets collected at varying sampling rates, all signals were resampled to 128 Hz and Z-score normalized to reduce inter-subject variability. For MCMED, we adopted the official split, whereas other datasets were randomly partitioned into 80% training and 20% testing at the subject level to prevent information leakage. Dataset details and final segment counts are reported in Appendix C.1.

**Baselines and Implementations.** To evaluate the effectiveness of our method, we compare it against representative GAN-based, diffusion-based, and flow-based generative models. Although these baselines were originally developed for short-duration PPG-to-ECG translation (e.g., 4 seconds), we extend them to the 10-second setting to ensure fair and consistent evaluation. All experiments were performed on a single NVIDIA A800 GPU with 80 GB memory. Implementation details are provided in Appendix C.2.

### 4.2 FIDELITY AND QUALITY ASSESSMENT OF SYNTHESIZED ECG

We evaluated the generated ECGs using Mean Absolute Error (MAE), Root Mean Square Error (RMSE), Fréchet Distance (FD), and Fréchet Inception Distance (FID). For FID, we used ECG-

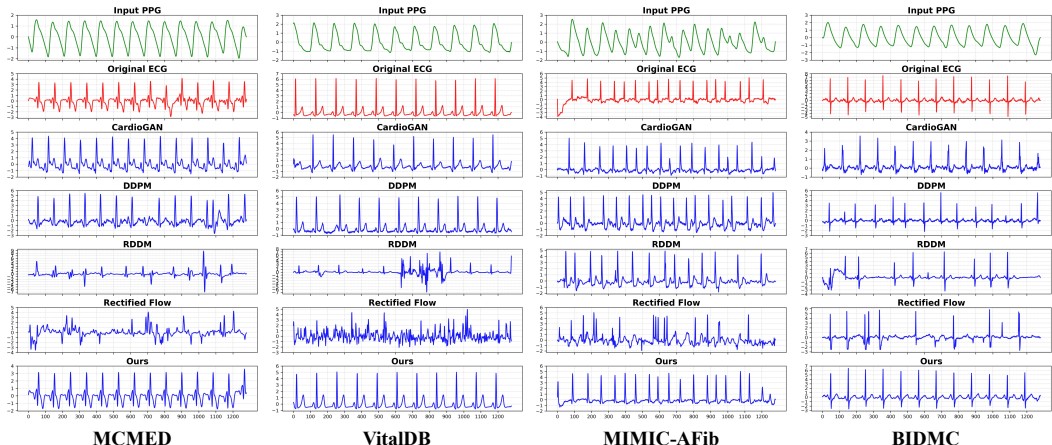

Figure 3: Qualitative comparison on generated ECG using varying nature of input PPG.

Founder (Li et al., 2024) as the feature extractor. We also estimated heart rate (in bpm) from both real and synthetic ECGs using Hamilton's method (Hamilton, 2002) and computed the corresponding heart rate error ($MAE_{HR}$). Because accurate heart rate estimation requires well-defined QRS complexes, $MAE_{HR}$ serves as a critical indicator of morphological fidelity.

The results in Table 1 demonstrate the effectiveness of our method. PPGFlowECG achieves state-of-the-art or highly competitive performance across all metrics and datasets, with consistent results across diverse clinical cohorts underscoring its robustness and generalizability. These gains arise from the synergy between the CardioAlign Encoder and Latent Rectified Flow: the encoder constructs a structured, semantically coherent latent space that provides a smooth manifold for the rectified flow to learn a stable vector field. Qualitative results in Figure 3 further highlight the superior fidelity of PPGFlowECG, particularly in clinically critical features such as QRS complexes and T-wave morphology that are essential for diagnosis. In addition, we evaluate improvements in heart rate estimation enabled by PPG-to-ECG translation (Appendix D.6) and provide external validation results in Appendix D.2. Collectively, these findings validate the efficacy of the framework and underscore its potential as a reliable tool for clinical-grade ECG synthesis from PPG signals.

### 4.3 CARDIOVASCULAR DISEASE DETECTION WITH SYNTHESIZED ECG

To assess the clinical utility of the synthesized signals, we evaluate their effectiveness in diagnosing a broad spectrum of cardiovascular diseases. As illustrated in Figure 4, the selected conditions test the diagnostic capacity of the generated signals across multiple pathophysiologies, extending beyond arrhythmias to include conduction, vascular, and structural disorders. On the MCMED dataset, we perform multi-label classification across six representative categories: atrial fibrillation and flutter (I48), aortic aneurysm and dissection (I71), atherosclerosis (I70), atrioventricular and left bundle-branch block (I44), chronic ischemic heart disease (I25), and heart failure (I50). This diverse spectrum challenges the model to capture subtle morphological features beyond rhythmic patterns. For this task, we adopt the Net-1d classifier (Hong et al., 2020) to ensure a fair comparison. We report AUROC for each category and Macro-AUROC as the overall measure of diagnostic performance. In addition, we conduct a binary classification task

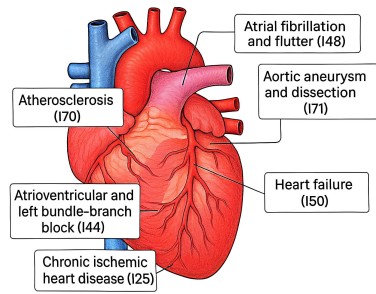

Figure 4: The evaluation encompasses cardiovascular diseases across rhythm, conduction, vascular, and structural domains.

on the MIMIC-AFib dataset to distinguish atrial fibrillation from normal rhythm, evaluating performance with accuracy, F1 score, and AUROC (see Appendix D.1 for results).

As shown in Table 2, PPGFlowECG achieves the highest overall performance in downstream classification tasks, with a Macro-AUROC of 0.631. In contrast, the standard rectified flow baseline,

Table 2: AUROC scores for multi-label cardiovascular disease classification on the MCMED dataset, with the Macro-AUROC reported as an overall measure of diagnostic performance.

| Methods | I48 | I71 | I70 | I44 | I25 | I50 | Macro-AUROC |
|---|---|---|---|---|---|---|---|
| CardioGAN (Sarkar & Etemad, 2021) | 0.612 | **0.657** | 0.567 | 0.570 | 0.574 | 0.539 | 0.587 |
| DDPM (T = 50) (Ho et al., 2020) | 0.666 | 0.605 | 0.576 | **0.625** | 0.584 | 0.591 | 0.608 |
| RDDM (T = 10) (Shome et al., 2024) | 0.574 | 0.576 | 0.538 | 0.573 | 0.544 | 0.525 | 0.555 |
| Rectified Flow (T = 10) (Liu et al., 2022) | 0.499 | 0.502 | 0.504 | 0.496 | 0.497 | 0.497 | 0.499 |
| **PPGFlowECG (T = 10)** | **0.708** | 0.626 | **0.622** | 0.619 | **0.608** | **0.604** | **0.631** |

which operates directly in the data space, reaches only 0.499—equivalent to random guessing. This highlights that our method's effectiveness stems not simply from adopting a flow-based architecture but from exploiting the structured and semantically coherent latent space constructed by the CardioAlign Encoder. By aligning modalities first, the framework establishes a smooth, physiologically meaningful manifold that enables stable vector field learning and ultimately yields reliable ECGs.

## 4.4 Ablation Study

As shown in Table 3 and Table 4, our progressive ablation study demonstrates that each component of PPGFlowECG contributes critically to overall performance. $\mathcal{L}_{\text{align}}$ provides the largest gain (Macro-AUROC 0.589→0.624) while improving signal fidelity, $\mathcal{L}_{\text{infoNCE}}$ sharpens the latent structure to boost classification (0.627), and $\mathcal{L}_{\text{cross}}$ delivers the best overall results (0.631) with the lowest reconstruction error. Additional ablation studies are presented in Appendix D.4 and Appendix D.5.

Table 3: Ablation study about signal quality metrics on MCMED.

| Ablation Options | MAE | RMSE | FD | FID | MAE$_{\text{HR}}$ |
|---|---|---|---|---|---|
| w/o CardioAlign Encoder | 0.78 | 1.22 | 34.85 | 9.84 | 2.12 |
| w CardioAlign Encoder ($\mathcal{L}_{\text{align}}$) | 0.74 | 1.16 | 43.79 | 10.11 | 1.84 |
| w CardioAlign Encoder ($\mathcal{L}_{\text{align}} + \mathcal{L}_{\text{infoNCE}}$) | 0.74 | 1.16 | 43.35 | 10.41 | 1.83 |
| w CardioAlign Encoder ($\mathcal{L}_{\text{align}} + \mathcal{L}_{\text{infoNCE}} + \mathcal{L}_{\text{cross}}$) | 0.73 | 1.14 | 43.99 | 12.84 | 1.80 |

Table 4: Ablation study about downstream diagnostic performance on MCMED.

| Ablation Options | I48 | I71 | I70 | I44 | I25 | I50 | Macro-AUROC |
|---|---|---|---|---|---|---|---|
| w/o CardioAlign Encoder | 0.620 | 0.578 | 0.593 | 0.601 | 0.571 | 0.574 | 0.589 |
| w CardioAlign Encoder ($\mathcal{L}_{\text{align}}$) | 0.709 | 0.607 | 0.625 | 0.606 | 0.598 | 0.599 | 0.624 |
| w CardioAlign Encoder ($\mathcal{L}_{\text{align}} + \mathcal{L}_{\text{infoNCE}}$) | 0.709 | 0.614 | 0.627 | 0.608 | 0.603 | 0.601 | 0.627 |
| w CardioAlign Encoder ($\mathcal{L}_{\text{align}} + \mathcal{L}_{\text{infoNCE}} + \mathcal{L}_{\text{cross}}$) | 0.708 | 0.626 | 0.622 | 0.619 | 0.608 | 0.604 | 0.631 |

## 4.5 Case Study and Explainability Analysis

We assess the clinical fidelity of the synthesized signals using Grad-CAM (Selvaraju et al., 2017), examining whether PPGFlowECG-generated ECGs preserve diagnostically relevant features. If clinically viable, their attention patterns closely mirror those of real ECGs. Figure 5 presents comparisons for two categories, showing strong correspondence in diagnostic attention patterns. This explainability analysis provides qualitative evidence that PPGFlowECG not only generates realistic waveforms but also reproduces subtle, clinically salient features essential for diagnosis, enabling downstream AI classifiers to rely on the same regions in both real and synthetic signals. Additional Grad-CAM visualizations are provided in Appendix D.7.

## 4.6 Clinical Interpretation and Evaluation

To assess the real-world applicability of our method, we conduct a two-part evaluation with certified cardiologists: (i) a clinical Turing Test to measure the perceptual realism of synthesized ECGs and (ii) an analysis of their diagnostic utility in atrial fibrillation detection. Five professional cardiologists participated, including one junior, three mediate, and one senior professionals.

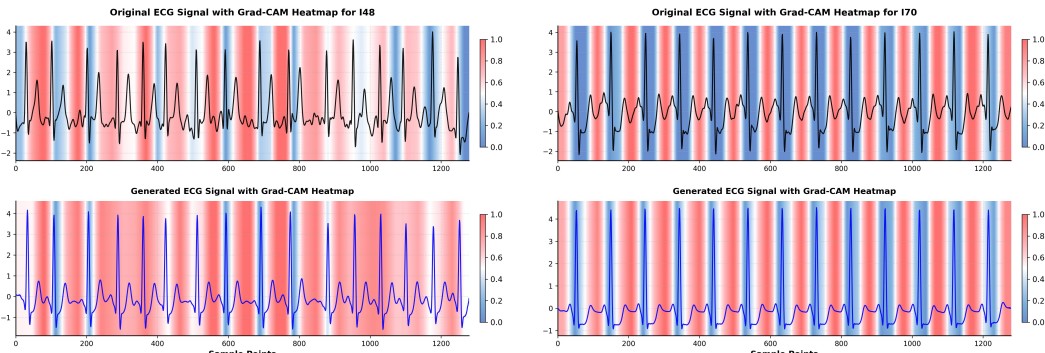

Figure 5: Explainability analysis with Grad-CAM on real (top) and synthesized (bottom) ECGs for two representative disease classes: I48 (left) and I70 (right).

**A Clinical Turing Test.** In the first task, cardiologists evaluate whether AI-generated ECGs can be distinguished from real recordings. Five certified cardiologists review a randomized sequence of 25 real and 25 synthesized signals, classifying each as either real or AI-generated. As shown in Table 5, the average classification accuracy is 0.52, only marginally above random chance (0.50). Strikingly, the average specificity is just 0.19, indicating that AI-generated signals are correctly identified only 19% of the time. This inability to reliably discriminate synthetic from real recordings highlights the high perceptual fidelity and clinical realism achieved by our method.

**Diagnostic Utility of Synthesized ECGs.** In the second task, five cardiologists diagnose atrial fibrillation from a randomized set of 50 signals (25 AF, 25 non-AF) under three data conditions. As presented in Table 6, PPG alone yields an average F1 score of 0.77. Adding real ECG as an auxiliary signal substantially improves performance to 0.93. Remarkably, replacing the real ECG with a synthesized ECG from PPGFlowECG not only preserves but further enhances diagnostic accuracy, achieving an F1 score of 0.94. These findings demonstrate that the generated signals convey diagnostic information equivalent to—and in this study even surpassing—real ECGs, underscoring the framework's potential as a reliable auxiliary tool in clinical settings where only PPG is available.

Table 5: Performance of five cardiologists on the Turing Test.

| Cardiologist ID | 1 | 2 | 3 | 4 | 5 | Avg |
|---|---|---|---|---|---|---|
| Accuracy | 0.44 | 0.70 | 0.50 | 0.50 | 0.44 | 0.52 |
| Specificity | 0 | 0.48 | 0.04 | 0.32 | 0.12 | 0.19 |

Table 6: Diagnostic performance of cardiologists for AF detection under various conditions.

| Test Modality | Accuracy | Sensitivity | Specificity | F1 Score |
|---|---|---|---|---|
| PPG | 0.84 | 0.70 | 0.98 | 0.77 |
| PPG & Real ECG | 0.93 | 0.90 | 0.97 | 0.93 |
| PPG & Gen.ECG (PPGFlowECG) | 0.93 | 0.96 | 0.90 | 0.94 |

## 5 CONCLUSION

In this paper, we propose PPGFlowECG, a novel two-stage framework that aligns PPG and ECG in a shared latent space through the CardioAlign Encoder and employs latent rectified flow to generate ECGs with high fidelity and clinical interpretability. By addressing challenges such as the misalignment of physiological semantics in generative models and the complexity of modeling in high-dimensional signals, the proposed approach establishes a new state of the art in both signal reconstruction quality and downstream diagnostic performance across clinical and real-world settings. These advances provide a strong foundation for cardiovascular screening and early detection in scenarios where only PPG is available, supporting the development of more reliable diagnostic tools in healthcare.

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

## A  STATEMENT ON THE USE OF LLM

During this research and manuscript preparation, we use Large Language Models (LLMs) for auxiliary tasks such as generating short scripts during coding and assisting with text translation and language polishing. All core ideas, research methodologies, and academic contributions are conceived and developed independently by the authors, with the role of LLMs limited to improving the fluency and readability of the presentation.

## B  ALGORITHM OF PPGFLOWECG

---

**Algorithm 1** Training in Stage 1: CardioAlign Encoder

---

1: **Input:** Dataset $\mathcal{D}$ of paired samples $(x_{\mathrm{ppg}}, x_{\mathrm{ecg}})$.
2: **Ensure:** Optimized parameters $\theta_{E_{\mathrm{CA}}}, \theta_{D_{\mathrm{PPG}}}, \theta_{D_{\mathrm{ECG}}}$.
3: **while** not converged **do**
4:      Sample a mini-batch $(x_{\mathrm{ppg}}, x_{\mathrm{ecg}}) \sim \mathcal{D}$.
5:      **for** $m \in \{\mathrm{ppg, ecg}\}$ **do**
6:          $(\boldsymbol{\mu}_m, \boldsymbol{\sigma}_m) \leftarrow E_{CA}(x_m)$
7:          $\mathbf{z}_m \leftarrow \boldsymbol{\mu}_m + \boldsymbol{\sigma}_m \odot \boldsymbol{\epsilon}$     where $\boldsymbol{\epsilon} \sim \mathcal{N}(\mathbf{0}, \mathbf{I})$
8:      **end for**
9:      Compute $\mathcal{L}_{\mathrm{rec}}, \mathcal{L}_{\mathrm{kl}}, \mathcal{L}_{\mathrm{align}}, \mathcal{L}_{\mathrm{infoNCE}}, \mathcal{L}_{\mathrm{cross}}$.
10:     $\mathcal{L}_{\mathrm{CardioAlign}}$ integrates $\mathcal{L}_{\mathrm{align}}, \mathcal{L}_{\mathrm{infoNCE}}$, and $\mathcal{L}_{\mathrm{cross}}$.
11:     $\mathcal{L}_{\mathrm{stage1}} \leftarrow \sum_{m \in \{\mathrm{ppg,ecg}\}} \left( \mathcal{L}_{\mathrm{rec}}^{(m)} + \alpha \mathcal{L}_{\mathrm{kl}}^{(m)} \right) + \mathcal{L}_{\mathrm{CardioAlign}}$     where $\alpha$ is set to $10^{-4}$
12:     Update $\theta_{E_{\mathrm{CA}}}, \theta_{D_{\mathrm{PPG}}}, \theta_{D_{\mathrm{ECG}}}$ using $\nabla \mathcal{L}_{\mathrm{stage1}}$
13: **end while**

---

**Algorithm 2** Training in Stage 2: Latent Rectified Flow

---

1: **Input:** Dataset $\mathcal{D}$, frozen encoder $E_{\mathrm{CA}}$, trainable flow model $v_\theta$.
2: **Ensure:** Optimized parameters $\theta$ for the flow model $v_\theta$.
3: **while** not converged **do**
4:      Sample a mini-batch $(x_{\mathrm{ppg}}, x_{\mathrm{ecg}}) \sim \mathcal{D}$.
5:      $c \leftarrow E_{\mathrm{CA}}(x_{\mathrm{ppg}})$
6:      $y \leftarrow E_{\mathrm{CA}}(x_{\mathrm{ecg}})$
7:      $x_t \leftarrow (1 - t)z + ty$     where $z \sim \mathcal{N}(\mathbf{0}, \mathbf{I})$, $t \sim \mathcal{U}(\mathbf{0}, \mathbf{1})$
8:      $\mathcal{L}_{\mathrm{stage2}} \leftarrow \| v_\theta(x_t, t, c) - (y - z) \|_2^2$
9:      Update $\theta$ using $\nabla \mathcal{L}_{\mathrm{stage2}}$
10: **end while**

---

**Algorithm 3** Sampling with $T$-steps

---

1: **Input:** Source PPG $x_{\mathrm{ppg}}$, number of steps $T$, frozen $E_{\mathrm{CA}}$, $D_{\mathrm{ECG}}$, trained flow model $v_\theta$.
2: **Output:** Synthetic ECG signal $\hat{x}_{\mathrm{ecg}}$.
3: $c \leftarrow E_{\mathrm{CA}}(x_{\mathrm{ppg}})$
4: $x_0 \sim \mathcal{N}(\mathbf{0}, \mathbf{I})$
5: $\Delta t \leftarrow 1/T$
6: **for** $k = 0, \ldots, T - 1$ **do**
7:      $t_k \leftarrow k \cdot \Delta t$
8:      $x_{k+1} \leftarrow x_k + \Delta t \cdot v_\theta(x_k, t_k, c)$
9: **end for**
10: $z_{\mathrm{ecg}} \leftarrow x_T$
11: $\hat{x}_{\mathrm{ecg}} \leftarrow D_{\mathrm{ECG}}(z_{\mathrm{ecg}})$

---

## C DETAILS IN EXPERIMENTAL SETTINGS

### C.1 DATASETS

Our framework is rigorously evaluated on four representative public datasets selected to cover diverse clinical scenarios, patient populations, and signal characteristics:

- **MCMED** (Kansal et al., 2025): The largest clinical-grade dataset of paired PPG–ECG signals, comprising recordings from 118,385 adult emergency department visits between 2020 and 2022. Its scale and expert-provided cardiovascular disease annotations make it uniquely valuable for diagnostic evaluation.

- **VitalDB** (Lee et al., 2022): A high-fidelity dataset of continuous PPG and ECG recordings from 6,388 non-cardiac surgical patients. It enables evaluation in perioperative settings but lacks explicit diagnostic labels.

- **MIMIC-AFib** (Bashar et al., 2019): A specialized arrhythmia dataset containing PPG and ECG recordings from 35 critically ill adults, with binary atrial fibrillation annotations derived from the MIMIC-III waveform database.

- **BIDMC** (Pimentel et al., 2016): Approximately 7 hours of synchronized PPG and multi-lead ECG recordings from 53 ICU patients. Although it lacks diagnostic labels, it is valuable for assessing robustness in intensive care settings.

To maintain a manageable test size for the largest datasets, we randomly sample subsets of test segments from MCMED and VitalDB. After standardized preprocessing, the final counts of high-quality 10-second PPG–ECG segment pairs used in our experiments are reported in Table 7.

Table 7: Dataset characteristics and counts of high-quality 10-second segments after preprocessing.

| Datasets | Orignal Sample Rate | Annotations | Training | Validation | Testing |
|---|---|---|---|---|---|
| MCMED | PPG 125Hz, ECG 500Hz | CVD labels | 10,356,840 | 1,299,869 | 37,052 |
| VitalDB | PPG 500Hz, ECG 500Hz | None | 1,007,165 | - | 10,720 |
| MIMIC-AFib | PPG 125Hz, ECG 125Hz | AF/none-AF | 2,044 | - | 577 |
| BIDMC | PPG 125Hz, ECG 125Hz | None | 1458 | - | 326 |

### C.2 IMPLEMENTATION DETAILS

**Training in Stage 1.** The first stage of our method consists of the shared CardioAlign Encoder and two modality-specific decoders for PPG and ECG reconstruction. The encoder employs a 1D CNN backbone with residual blocks to capture local waveform features, complemented by a lightweight self-attention layer for modeling long-range temporal dependencies. It applies five successive $\times 2$ temporal downsampling operations, resulting in an overall $\times 32$ reduction. The latent head outputs the posterior distribution parameters, $\boldsymbol{\mu}, \boldsymbol{\sigma} \in \mathbb{R}^{L/32 \times 4}$, from which the latent representation $\mathbf{z}$ is sampled using the reparameterization trick and scaled by 0.18215 to stabilize the latent space. The decoder follows a hybrid design that reconstructs both high-frequency and low-frequency components: a residual upsampling path recovers fine-grained details, while a TimeVAE-inspired (Desai et al., 2021) decomposable head models the global structure through level, polynomial trend, and seasonal components, which are summed to form the final signal. Stage 1 is trained for 40,000 iterations with a batch size of 128 using the AdamW optimizer with an initial learning rate of $2 \times 10^{-5}$.

**Training in Stage 2.** On top of the frozen Stage 1, we train a conditional latent rectified flow to deterministically map PPG latents to ECG latents. For each paired input, $\mathbf{z}_{\mathrm{ppg}}$ and $\mathbf{z}_{\mathrm{ecg}}$ are extracted using the frozen CardioAlign Encoder, and the latent rectified flow is optimized with $target = \mathbf{z}_{\mathrm{ecg}}$, and $cond = \mathbf{z}_{\mathrm{ppg}}$. Stage 2 is trained with the Adam optimizer (learning rate $1 \times 10^{-4}$), gradient clipping at 1.0, and an exponential moving average (EMA) of parameters with decay 0.995, updated every 10 steps.

# D  ADDITIONAL EXPERIMENTS AND RESULTS

## D.1  ATRIAL FIBRILLATION DETECTION ON MIMIC-AFIB

We adopt a VGG-13 classifier (Sallem et al., 2020) for a fair comparison and evaluate atrial fibrillation detection on the MIMIC-AFib dataset as a binary classification task, using accuracy, F1 score, and AUROC as performance metrics. Atrial fibrillation is a common arrhythmia for which accurate diagnosis is essential to preventing severe complications such as stroke. As reported in Table 8, PPGFlowECG outperforms all prior state-of-the-art methods, achieving an accuracy of 0.82, an F1 score of 0.87, and an AUROC of 0.87. These results demonstrate that the ECGs synthesized by PPGFlowECG are both realistic and clinically informative, underscoring the framework's potential for diagnostic applications.

Table 8: Evaluation of atrial fibrillation detection on MIMIC-AFib.

| Methods | Accuracy | F1 Score | AUROC |
|---|---|---|---|
| CardioGAN (Sarkar & Etemad, 2021) | 0.71 | 0.83 | 0.34 |
| DDPM (T = 50) (Ho et al., 2020) | 0.70 | 0.80 | 0.58 |
| RDDM (T = 10) (Shome et al., 2024) | 0.70 | 0.82 | 0.41 |
| Rectified Flow (T = 10) (Liu et al., 2022) | 0.61 | 0.71 | 0.52 |
| **PPGFlowECG (T = 10)** | **0.82** | **0.87** | **0.87** |

## D.2  EXTERNAL VALIDATION OF GENERALIZABILITY

To rigorously evaluate the generalization capacity of our framework, we conduct an external validation study of zero-shot synthesis. The objective is to test whether a generalist model, trained on a large and diverse dataset, can generate high-quality ECGs for a distinct, unseen domain. We compare two settings: **(i) Internal Validation (Specialist Model)**: trained and evaluated exclusively on the MIMIC-AFib dataset, providing a domain-specific benchmark; and **(ii) External Validation (Generalist Model)**: trained solely on the large-scale MCMED dataset and applied directly, without fine-tuning, to the MIMIC-AFib dataset for ECG synthesis and evaluation.

**Analysis of Signal-Level Metrics.** As shown in Table 9, the generalist model shows modest degradation in waveform fidelity (MAE, RMSE) but greater errors in heart rate estimation ($MAE_{HR}$), underscoring the difficulty of capturing fine morphological details in a zero-shot setting. Distributional metrics yield mixed results, with FID worsening as expected but FD paradoxically improving, likely due to domain-specific signal characteristics. These findings highlight the ambiguity of signal-level metrics and underscore the importance of downstream diagnostic performance as the definitive measure of clinical utility.

Table 9: Signal-level metrics on MIMIC-AFib under internal and external validation. For all metrics, lower values indicate better performance. The best result for each metric is **bolded**.

| Methods | MAE | RMSE | FD | FID | $MAE_{HR}$ |
|---|---|---|---|---|---|
| Internal Validation (Specialist Model) | **0.73** | **1.29** | 63.75 | **37.69** | **2.30** |
| External Validation (Generalist Model) | 0.84 | 1.31 | **11.47** | 73.17 | 5.39 |

**Analysis of Downstream Diagnostic Performance.** As shown in Table 10, performance on the AF detection task provides a definitive measure of generalization. The specialist model (internal validation) achieves an F1 score of 0.87, while the generalist model (external validation), despite never being trained on MIMIC-AFib, maintains strong diagnostic accuracy with an F1 Score of 0.82 and an AUROC of 0.83. This robust downstream performance resolves the ambiguity observed in signal-level metrics, demonstrating that the framework preserves and transfers the pathological features essential for diagnosis. These results validate the framework's powerful zero-shot generalization capability and highlight its potential for applications where paired target-domain training data are unavailable.

Table 10: Diagnostic performance on MIMIC-AFib under internal and external validation.

| Methods | Accuracy | F1 Score | AUROC |
|---|---|---|---|
| Internal Validation (Specialist Model) | **0.82** | **0.87** | **0.87** |
| External Validation (Generalist Model) | 0.77 | 0.82 | 0.83 |

## D.3 PER-CATEGORY ROC CURVES FOR MULTI-LABEL CLASSIFICATION ON MCMED

To provide a more detailed view of downstream diagnostic performance, we present per-category Receiver Operating Characteristic (ROC) curves for the multi-label classification task on the MCMED dataset. This task covers six distinct cardiovascular disease categories, each posing unique diagnostic challenges. For each category, the ROC curve illustrates the trade-off between true-positive and false-positive rates, with the Area Under the Curve (AUROC) quantifying performance. Figure 6 shows the ROC curves for all six ICD-10 categories, comparing PPGFlowECG against baseline methods. The results corroborate the summary findings reported in the main text, highlighting our model's consistent superiority. Notably, the standard Rectified Flow baseline frequently lies close to the random-chance diagonal (AUROC ≈ 0.5), underscoring the diagnostic reliability of our synthesized ECGs across diverse cardiovascular conditions.

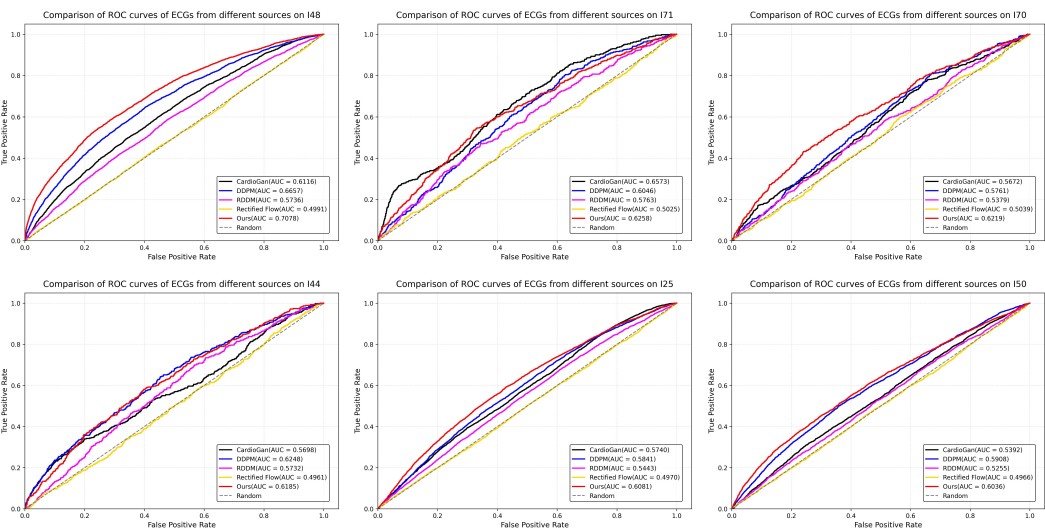

Figure 6: Per-category Receiver Operating Characteristic (ROC) curves for multi-label cardiovascular disease classification on the MCMED dataset.

## D.4 VISUALIZATION OF LATENT SPACE ALIGNMENT

To illustrate the effectiveness of the CardioAlign Encoder in learning a shared, modality-invariant latent space, we apply t-Distributed Stochastic Neighbor Embedding (t-SNE) (Maaten & Hinton, 2008) to project high-dimensional ECG and PPG representations into three dimensions. We compare our shared encoder with a baseline model that employs two independent encoders without alignment. As shown in Figure 7, the baseline (left) produces distinct clusters for ECG (blue) and PPG (orange), indicating a substantial modality gap. In contrast, the CardioAlign Encoder (right) yields intermingled distributions, forming a unified latent space where the two modalities are indistinguishable. This visualization demonstrates that the shared encoder, reinforced by multi-level alignment objectives described in the main text, effectively bridges the modality gap, a critical factor in generating high-fidelity cross-modal signals.

## D.5 ANALYSIS OF SAMPLING STEPS

To evaluate the effect of the number of ODE solver steps ($T$) on generation quality, we conduct an ablation study varying $T$ from 5 to 25. As shown in Table 11, increasing the number of steps gener-

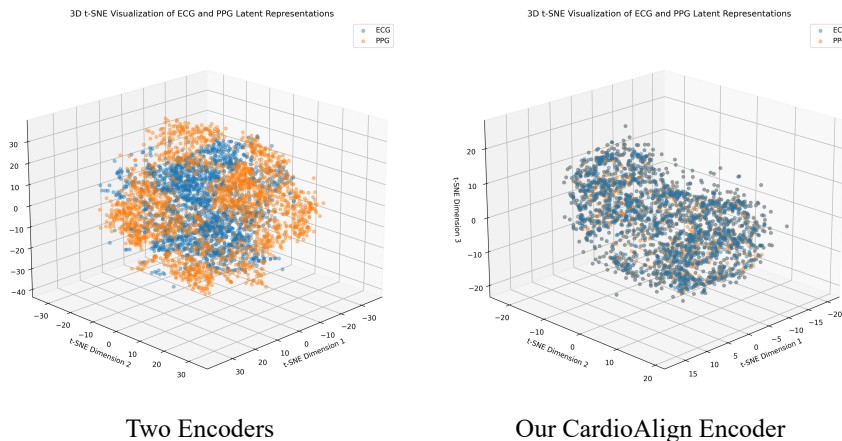

Two Encoders                    Our CardioAlign Encoder

Figure 7: 3D t-SNE visualizations of ECG and PPG latent representations.

ally improves distributional metrics (FD and FID). However, metrics directly reflecting per-sample waveform fidelity and physiological accuracy, such as MAE, RMSE, and MAE$_{HR}$—are optimal at just $T = 5$. This observation is corroborated by the downstream diagnostic performance in Table 12, where the model achieves its highest Macro-AUROC of 0.640 at 5 steps. Diagnostic accuracy then declines steadily as the number of steps increases. These findings underscore the exceptional efficiency of our framework, which generates high-fidelity, clinically informative signals with minimal computational cost.

For the comparative experiments in the main text, we fix $T = 10$ to ensure a fair comparison with baseline models (e.g., RDDM), which also adopt 10 steps in their original configurations. Notably, even at this slightly suboptimal setting for our method, PPGFlowECG achieves state-of-the-art performance, while the ablation confirms its efficiency advantage in practical applications.

Table 11: Ablation study of the effect of sampling steps on signal quality metrics.

| Sampling Steps | MAE | RMSE | FD | FID | MAE$_{HR}$ |
|---|---|---|---|---|---|
| $T = 5$ | 0.71 | 1.12 | 54.22 | 16.57 | 1.78 |
| $T = 10$ | 0.73 | 1.14 | 43.99 | 12.84 | 1.80 |
| $T = 15$ | 0.74 | 1.15 | 39.99 | 11.46 | 1.87 |
| $T = 20$ | 0.74 | 1.16 | 37.77 | 10.75 | 1.88 |
| $T = 25$ | 0.75 | 1.16 | 36.38 | 10.33 | 1.89 |

Table 12: Ablation study of the effect of sampling steps on downstream diagnostic performance.

| Sampling Steps | I48 | I71 | I70 | I44 | I25 | I50 | Macro-AUROC |
|---|---|---|---|---|---|---|---|
| $T = 5$ | 0.718 | 0.633 | 0.632 | 0.625 | 0.617 | 0.613 | 0.640 |
| $T = 10$ | 0.708 | 0.626 | 0.622 | 0.619 | 0.608 | 0.604 | 0.631 |
| $T = 15$ | 0.704 | 0.624 | 0.621 | 0.615 | 0.605 | 0.600 | 0.628 |
| $T = 20$ | 0.702 | 0.622 | 0.620 | 0.613 | 0.603 | 0.598 | 0.626 |
| $T = 25$ | 0.701 | 0.620 | 0.618 | 0.612 | 0.602 | 0.597 | 0.625 |

## D.6 IMPROVEMENT IN HEART RATE ESTIMATION

To further assess the physiological fidelity of the synthesized signals, we evaluate whether PPG-to-ECG translation enhances heart rate (HR) estimation. We compare HR derived directly from the source PPG with HR estimated from the corresponding ECG synthesized by PPGFlowECG, using HR from the real ECG as the ground truth. As shown in Figure 8, HR estimates from synthesized ECGs align more closely with the $y = x$ identity line, indicating stronger agreement with the ground truth, whereas PPG-based estimates exhibit greater variance and more frequent outliers, particularly at extreme heart rates. Quantitatively, synthesized ECGs reduce the Mean Absolute Error (MAE)

from 2.16 to 1.80 BPM, demonstrating that PPG-to-ECG translation not only preserves but also refines cardiac rhythm information. These findings highlight the value of synthesized ECGs as a more robust source for downstream physiological parameter estimation, reinforcing the clinical relevance of our framework.

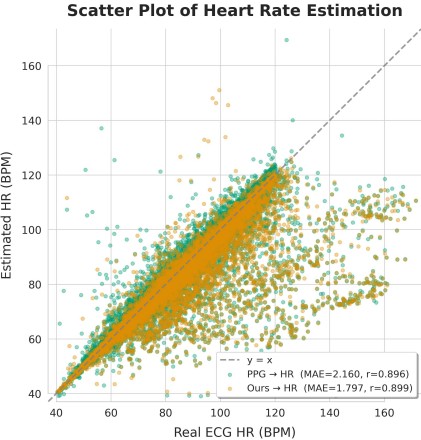

Figure 8: Scatter plot comparing Heart Rate (HR) estimation accuracy.

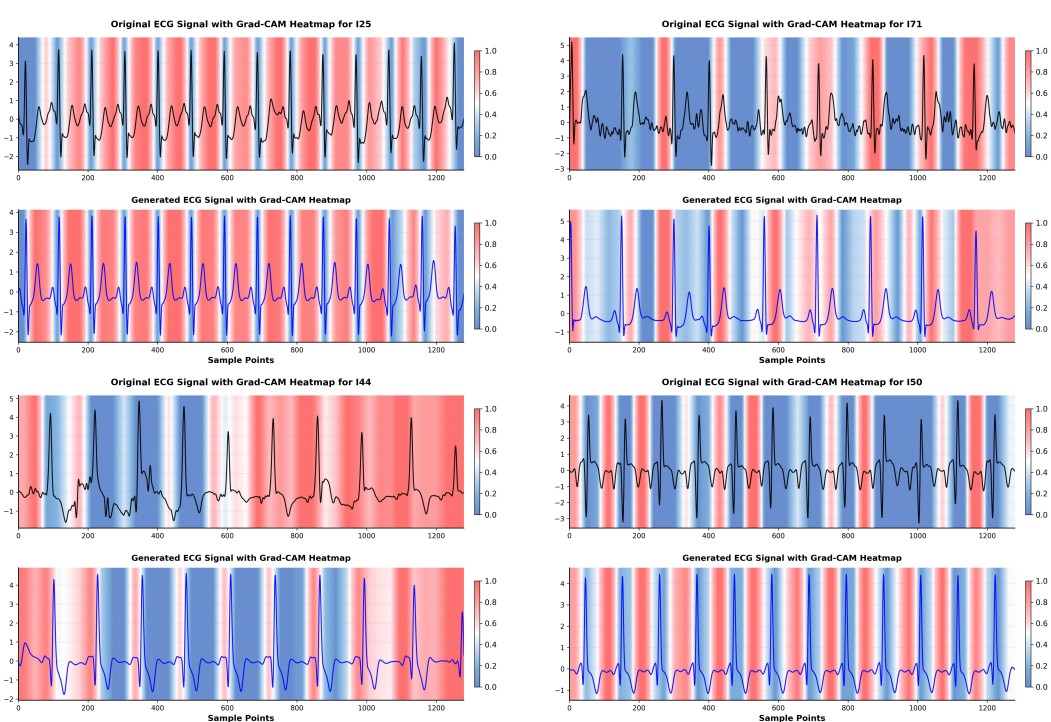

Figure 9: Grad-CAM visualizations for the remaining four cardiovascular disease categories.

## D.7 ADDITIONAL GRAD-CAM VISUALIZATIONS FOR MCMED DISEASE CLASSES

To extend the explainability analysis presented in the main text, we provide the complete set of Grad-CAM visualizations for the remaining four cardiovascular disease categories in the MCMED dataset. As discussed in the main paper, this analysis qualitatively verifies that signals generated by PPGFlowECG preserve the diagnostically relevant features utilized by downstream classifiers in real ECGs. Figure 9 presents additional comparisons for chronic ischemic heart disease (I25), aortic aneurysm and dissection (I71), Atrioventricular and left bundle-branch block (I44), and heart failure

(I50). Consistent with the findings in the main body, these results show strong correspondence in diagnostic attention patterns between real ECGs (top row) and their synthesized counterparts (bottom row). These supplementary visualizations further substantiate that PPGFlowECG faithfully reproduces subtle, clinically salient features essential for diagnosis. The strong alignment of Grad-CAM heatmaps provides compelling visual evidence that the generative process is semantically and diagnostically coherent, producing signals genuinely useful for clinical applications.

# E DISCUSSION ON LIMITATIONS AND FUTURE WORKS

## E.1 ANALYSIS OF GENERATION FAILURE CASES

A key limitation of the current framework is its sensitivity to input PPG quality. The main experiments are conducted on pre-processed, high-quality segments to establish benchmark performance under ideal conditions. However, performance degrades markedly when presented with corrupted inputs, which are common in real-world ambulatory settings. Figure 10 illustrates two representative failure cases in which the model generates ECGs from low-quality PPG signals containing severe motion artifacts. In both examples, the input PPG (top row) is highly irregular and loses its characteristic pulsatile morphology, leading the synthesized ECG (bottom row) to deviate substantially from the ground-truth ECG (middle row). Although the model preserves a rhythmic pattern, its fidelity is significantly hindered by corrupted inputs. This limitation underscores the importance of input quality and motivates future work to improve robustness, either by training on more diverse datasets that include noisy signals or by integrating advanced signal processing and noise-aware mechanisms into the model architecture.

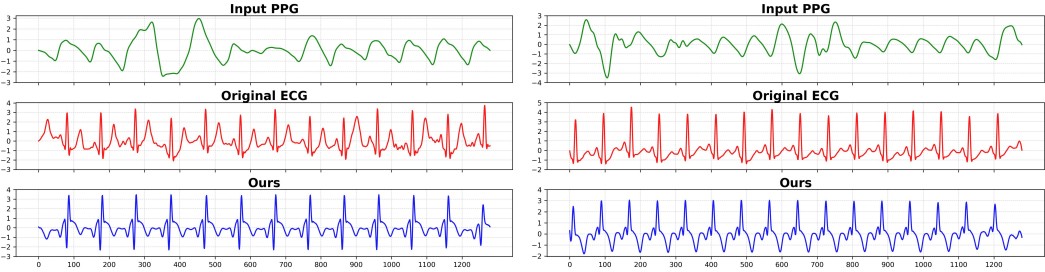

Figure 10: Failure cases illustrating the model's sensitivity to degraded PPG inputs.

## E.2 FUTURE WORKS

**Robustness to Low-Quality PPG Signals.** Our framework currently relies on a preprocessing pipeline that discards low-quality signal segments to ensure reliable input. However, in ambulatory settings, wearable PPG signals are often degraded by motion artifacts or sensor displacement. The robustness of the model under such conditions remains insufficiently assessed. Future work should explore integrating advanced signal quality assessment and noise-robust representation learning into the CardioAlign Encoder to enhance performance on uncurated, real-world data.

**Scope of Cardiovascular Conditions.** Our downstream evaluation focuses on six cardiovascular diseases from the MCMED dataset, which span multiple pathophysiological domains but do not cover the full range of ECG-diagnosable disorders, such as complex non-AFib arrhythmias, specific myocardial infarction patterns, or channelopathies. Future work should expand evaluation to these conditions to determine the framework's capacity to synthesize their distinctive and subtle morphological features.

**Extension to Multi-Lead ECG Generation.** The current framework is restricted to generating a single-lead (Lead II) ECG. While Lead II is highly informative for rhythm assessment, the clinical gold standard for comprehensive diagnosis is the 12-lead ECG, which captures multiple complementary perspectives on cardiac electrical activity. Extending PPGFlowECG to synthesize a full 12-lead ECG from a single PPG input would greatly enhance its clinical utility but presents substantial challenges in accurately modeling inter-lead correlations.

