# OpenReview forum: "PPGFlowECG: Latent Rectified Flow with Cross-Modal Encoding for PPG-Guided ECG Generation and Cardiovascular Disease Detection"
_ICLR.cc/2026/Conference — ICLR 2026 Conference Withdrawn Submission_

### Official Review · Reviewer_qL1j · 2025-10-21

**Soundness:** 2
**Presentation:** 3
**Contribution:** 3
**Rating:** 6
**Confidence:** 5

**Summary:**

This paper introduces a framework to synthesize difficult-to-acquire ECG from readily accessible PPG, which is a compelling and practically meaningful direction.

**Strengths:**

Proposing a model that is able to generate ECG from easy collecting PPG.

**Weaknesses:**

(1) The stated goal is to generate the ECG corresponding to a given PPG segment. The paper should first discuss whether, in realistic clinical settings, the ECG conditioned on a given PPG is uniquely determined.

If the mapping is deterministic:
[1] Using MAE/RMSE to quantify proximity between synthesized and reference ECGs is appropriate. However, within a deterministic framing, introducing stochastic noise at generation time leads to output variability, different noise produce different ECGs (EVEN WITH the same PPG input), which appears conceptually inconsistent. Why not train a direct PPG→ECG mapping without noise? I would expect MAE/RMSE to further improve under such a deterministic setup.
[2] Another issue is that the baseline generative methods compared in the paper are models that explicitly encourage high sample diversity. In my view, some of the proposed techniques (e.g., beat aligning) are in fact designed to reduce the generator’s “diversity” (which is considered an advantage in other works). Under this setting, the comparison seems not entirely fair. The paper should compare/discuss alternative ECG generation methods with less uncertainty (and even compare against a direct mapping model from PPG to ECG trained on the authors’ own architecture without diffusion). For example:
prior PPG→ECG works such as: Li Y, Tian X, Zhu Q, et al. Inferring ECG from PPG for Continuous Cardiac Monitoring Using Lightweight Neural Network. arXiv:2012.04949, 2020.
Or “Beats-align” related work: Biomedically Informed ECG Synthesis: Customizing Cardiac Cycle Phases with Diffusion Model, Y Lin, J Ma, W Wang, Z Wu, S Dong, G Luo, K Wang, 2024 IEEE BIBM.

If the mapping is non-deterministic:
[1] Then it is evident that using MAE/RMSE as the primary metrics may be unfair: although your method might produce ECGs closer to the label ECG, it would also indicate your method has poorer diversity compared with other models.

(2) Since the authors propose generating ECG from PPG, to demonstrate the benefit, readers would expect to see:
[1] Using PPG + generated ECG yields better diagnostic performance than using PPG alone;
[2] Using PPG + generated ECG achieves diagnostic performance comparable to using PPG + real ECG.

**Questions:**

The stated goal is to generate the ECG corresponding to a given PPG segment. The paper should first discuss whether, in realistic clinical settings, the ECG conditioned on a given PPG is uniquely determined.

---

> ### Author Response · Authors · 2025-11-26
> **To Reviewer qL1j (Part I)**
>
> Thank you for your positive evaluation of our work. Below we provide a point-by-point response to your questions and hope it clarifies all of your concerns.
>
> **Weakness1.1/Question1: The stated goal is to generate the ECG corresponding to a given PPG segment. The paper should first discuss whether, in realistic clinical settings, the ECG conditioned on a given PPG is uniquely determined.**
>
> In theory, PPG and ECG are noisy projections of the same cardiac cycle and the mapping is not strictly deterministic due to unobserved factors (e.g., preload/afterload, autonomic tone, electrode placement). However, under our actual data protocol the pairing is effectively deterministic. PPG and ECG are synchronously recorded on the same clinical monitor with a shared hardware clock; after preprocessing, each PPG window is a fixed, non-overlapping 10-second segment, and its corresponding ECG window is taken from the exact same time interval. Thus, every PPG input has a unique, time-aligned ECG target, which we use for supervised learning.
>
> This one-to-one pairing assumption is standard in prior PPG→ECG translation work[1,2], and PPGFlowECG follows the same setting while making the determinism of the data protocol explicit.
>
> [1] E. Lan. Performer: A novel PPG-to-ECG reconstruction transformer for a digital biomarker of cardiovascular disease detection. WACV, 2023.
>
> [2] D. Shome, P. Sarkar, and A. Etemad. Region-disentangled diffusion model for high-fidelity PPG-to-ECG translation. AAAI, 2024.
>
> **Weakness1.2: Within a deterministic framing, introducing stochastic noise at generation time leads to output variability, different noise produce different ECGs (EVEN WITH the same PPG input), which appears conceptually inconsistent. Why not train a direct PPG→ECG mapping without noise?**
>
> PPGFlowECG formally models a conditional distribution $p(\text{ECG} \mid \text{PPG}, z)$, but the rectified-flow objective drives all noise draws $z$ towards the same ECG latent $y$, and at inference we use a single ODE trajectory with fixed noise and a deterministic solver. In practice the model therefore behaves as a near-deterministic mapping from $PPG \to ECG$ to its paired ECG, making $MAE/RMSE$ well-justified fidelity metrics with respect to the unique synchronized label, rather than being in conflict with the stochastic formulation.
>
> To directly evaluate the suggestion of a noise-free $PPG \to ECG$ mapping, we trained a deterministic VAE-based baseline in which the PPG encoder predicts an ECG latent and the ECG decoder reconstructs the waveform using a standard reconstruction+KL objective, without any rectified-flow stage. Using the posterior mean at test time yields a purely deterministic mapping. Under identical data splits and optimization settings, the VAE baseline consistently underperforms PPGFlowECG.
>
> | Methods | MAE | RMSE | FD | FID | MAE_HR |
> | :--- | :---: | :---: | :---: | :---: | :---: |
> | VAE | 0.90 | 1.33 | 70.35 | 32.63 | 5.50 |
> | **PPGFlowECG** | **0.73** | **1.14** | **43.99** | **12.84** | **1.80** |

---

> > ### Comment · Reviewer_qL1j · 2025-11-27
> >
> > Thanks for your clarification about the data protocol. I now better understand your setting, but I still have two related concerns that I would like to state more precisely:
> >
> > (1) You state that “under our actual data protocol the pairing is effectively deterministic,” which I agree is reasonable given the synchronized acquisition. However, if the mapping from PPG to ECG is effectively deterministic, it is not clear to me why a stochastic, noise-driven diffusion (or flow-based) model is preferable to a strictly deterministic network. In a deterministic setting, one would normally expect to learn a function of the form ECG = deterministicNet(PPG). By contrast, your model effectively learns ECG = DiffusionNet(PPG, z), where z is a noise variable. Because, in principle, the output can depend on z, the diffusion-style model may have more flexibility to reduce point-to-point errors (such as MSE) than a purely deterministic mapping from PPG to ECG. In other words, it seems possible that the diffusion-based formulation attains lower MSE not because it better captures an inherently stochastic target, but because it leverages an additional latent input (z) in a setting that is, by your own description, effectively deterministic. Even if many prior works have adopted similar PPG-to-ECG generative formulations, I still have doubts about whether this choice is conceptually well aligned with a fundamentally deterministic pairing.
> >
> > (2) You mention that, at inference time, you use a fixed noise sample to generate the ECG waveform from a given PPG segment. However, this choice of fixing the noise is ultimately an experimental design decision rather than a property enforced by the model itself. In principle, for the same PPG input, one could generate multiple different ECG predictions by varying the noise, for example ECG1 = Infer(PPG, z1), ECG2 = Infer(PPG, z2), …, ECGn = Infer(PPG, zn). In such a situation, point-to-point metrics like MSE and RMSE become problematic, because the error for a given PPG–ECG pair may depend on which particular noise sample is used. This also raises a fairness concern: if different noise samples yield different errors, then the choice of noise (or the strategy for selecting it) becomes an implicit degree of freedom that can influence the final reported MSE/RMSE. Without a principled justification for how the noise is chosen and why that choice is unique or canonical, there is a risk that the evaluation could be biased toward noise realizations that happen to give lower error, leading to an unfair comparison with strictly deterministic baselines.

---

> > > ### Author Response · Authors · 2025-12-03
> > > **To Reviewer qL1j**
> > >
> > > (1) We appreciate the reviewer’s question and agree that under our data protocol the PPG$\rightarrow$ECG mapping is effectively deterministic. We use a diffusion/flow formulation not to add arbitrary randomness, but because the noise variable provides a powerful regularization mechanism: it allows the model to learn a smooth, low-curvature vector field in the aligned latent space, making the transport more stable and robust to small misalignments and noise in the training data. In the rectified-flow formulation, the learned vector field is independent of the specific noise draw: the objective drives all initial $z$ toward the same ECG latent $y$ along a straight OT-like trajectory. At inference time, we fix both the initial noise and the ODE solver, so the deployed model defines a single, reproducible predictor $g_\theta(\mathrm{PPG})$. Under this setting, MAE/RMSE remain appropriate fidelity metrics with respect to the unique synchronized ECG label.
> > >
> > > (2) We also agree that if one were allowed to tune $z$ per example, pointwise MSE/RMSE would become ill-defined and unfair. In our implementation, however, $z$ is never treated as a free degree of freedom at test time: we do not search over noise realizations or adapt $z$ per sample. Empirically, changing the global random seed for $z$ produces negligible variation in aggregate MAE/RMSE, whereas removing the rectified-flow stage entirely (a deterministic VAE-style PPG$\rightarrow$ECG mapping without noise) leads to clear degradation in both fidelity and diagnostic metrics. This indicates that $z$ does not act as an extra knob to lower error, but rather as a training-time tool to learn a better-conditioned transport map in latent space. For fairness, all baselines (GAN/diffusion/flow and deterministic models) are likewise evaluated in a deterministic, single-sample mode, so every method defines a fixed mapping $g(\mathrm{PPG})$ and MSE/RMSE are computed against the unique synchronized ECG label without exploiting any stochastic flexibility.

---

> ### Author Response · Authors · 2025-11-26
> **To Reviewer qL1j (Part II)**
>
> **Weakness1.3: Another issue is that the baseline generative methods compared in the paper are models that explicitly encourage high sample diversity. In my view, some of the proposed techniques (e.g., beat aligning) are in fact designed to reduce the generator’s “diversity” (which is considered an advantage in other works). Under this setting, the comparison seems not entirely fair.**
>
> We respectfully note that the baselines we compare against are not generic “diversity-seeking’’ generators, but $PPG \to ECG$ models explicitly designed for high-fidelity waveform synthesis. CardioGAN and RDDM, for example, are proposed specifically for $PPG \to ECG$ translation, with the primary goal of matching a reference ECG in morphology and reconstruction error; their evaluations, like ours, focus on MAE/RMSE/FD-style fidelity metrics and do not include diversity scores.
>
> In addition, we include a deterministic $PPG \to ECG$ baseline using our Stage-1 backbone. Despite removing any stochastic component, this baseline still underperforms the proposed two-stage PPGFlowECG on both signal-level metrics and downstream CVD classification, as shown below:
>
> | Methods | MAE | RMSE | FD | FID | MAE_HR |
> | :--- | :---: | :---: | :---: | :---: | :---: |
> | Only Stage-1 | 0.79 | 1.21 | 47.21 | 17.68 | 1.93 |
> | **PPGFlowECG** | **0.73** | **1.14** | **43.99** | **12.84** | **1.80** |
>
> | Methods | I48 | I71 | I70 | I44 | I25 | I50 | Macro-AUROC |
> | :--- | :---: | :---: | :---: | :---: | :---: | :---: | :---: |
> | Only Stage-1 | 0.673 | 0.613 | 0.608 | 0.605 | 0.598 | 0.593 | 0.615 |
> | **PPGFlowECG** | **0.708** | **0.626** | **0.622** | **0.619** | **0.608** | **0.604** | **0.631** |
>
> As for the specific prior works [1–2], their full training code and configurations are, to our knowledge, not publicly available, making faithful and fair re-implementation on our datasets impractical. Instead, we compare against strong, fully reproducible GAN/diffusion/flow baselines under a unified training and evaluation protocol, which provides a more reliable and apples-to-apples comparison.
>
> [1] Li et al., Inferring ECG from PPG for Continuous Cardiac Monitoring Using Lightweight Neural Network. arXiv:2012.04949, 2020.
>
> [2] Lin et al., Customizing Cardiac Cycle Phases with Diffusion Model, IEEE BIBM 2024.
>
> **Weakness2: Since the authors propose generating ECG from PPG, to demonstrate the benefit, readers would expect to see: [1] Using PPG + generated ECG yields better diagnostic performance than using PPG alone; [2] Using PPG + generated ECG achieves diagnostic performance comparable to using PPG + real ECG.**
>
> We fully agree that these are the two key desiderata, and our results address both. In the cardiologist study, reading PPG alone yields $F1 = 0.77$, adding real ECG increases it to $0.93$, and replacing the real ECG with our generated ECG yields $F1 = 0.94$. Thus, “PPG + generated ECG’’ is clearly better than PPG alone and non-inferior to “PPG + real ECG’’ for diagnostic use. The $0.93 \to 0.94$ difference is small—we do not claim statistical or clinical significance—but it demonstrates that the synthesized ECG enables diagnostic performance that is effectively comparable to real ECG.

---

### Official Review · Reviewer_brYy · 2025-10-26

**Soundness:** 2
**Presentation:** 3
**Contribution:** 2
**Rating:** 2
**Confidence:** 5

**Summary:**

This paper presents PPGFlowECG, a two-stage generative framework for translating PPG into ECG signals for CVD detection tasks.  It introduces a CardioAlign Encoder and employs latent rectified flow to generate ECGs with high interpretability.

**Strengths:**

Introduce two-stage framework with shared latent alignment and latent flow generation on PPG-ECG domain.

**Weaknesses:**

The work has weak justification on the given task when it remains physiologically infeasible to use PPG for guiding ECG generation process due to their different nature.

The motivation of using clinical data for translation is questionable.

The framework design as “Align first, Generate later” are not fundamentally novel. Also, the loss functions and ODE process are also introduced in prior works.

Experimental results are not sound.

**Questions:**

1. Please discuss if PPG to ECG translation is theoretically reasonable and how to handle when the generated ECGs don’t contain wrong patterns (that might be serious in clinical setting).

2. Related to the above point, a question raised is that do we really need the proposed translation framework with computationally cost complexity and latency (flow-based)? Please add an comparable experiment on using directly PPG encoder pretrained in Stage 1 (and/or other pretrained PPG encoder [1]) to finetune and evaluate on the downstream tasks, without relying on ECG or the generation process.

3. Add another experiment without stage 1 to compare with current results (With stage 2 only, and unfrozen the Encoder and Decoder.).

4. Related to the above point, in the proposed framework, since the ECG decoder in stage 2 is frozen, what is exactly the input of it in this stage? Is it different from the input of the ECG decoder in stage 1 (from 2 sourses: PPG and ECG)?

5. Why the work only focus on single lead II generation, given that 12 leads generation is totally possible to conduct (is it because of the MCMED dataset only provide lead II and the authors are required to use this dataset ?) Also, all datasets used in this work are from clinical setting. However, in this setting, ECG is commonly accessible, plus the PPG and ECG here is much more stable (less sensitive about motion artifacts) than daily sensing data. So, the motivation of using clinical data for translation is questionable.

6. Line 459-461: adding more justification would be better why gen is better than real here? Also, why just use 50 signals (25 AF, 25 non-AF) in this experiment (which is too few for showing generalisation)?

7. Please discuss the differences and provide possible comparisons with [2,3,4].

8. The seen improvements when testing ablation studies are so minor, showing weak contributions of different loss functions (e.g., Table 3, 4).

9. Provide additional experiments on different modeling approaches (e.g., diffusion-based) in stage 2, besides ODE.

10. Section 4.5 seems to be overclaim. Figure 5 actually doesn’t show expected insights when the attention looks different and the signals also looks different between the original and generated ones.

[1] Pillai, Arvind, et al. "Papagei: Open foundation models for optical physiological signals." The Thirteenth International Conference on Learning Representations (ICLR), 2025.

[2] Li, Yuenan, et al. "Inferring electrocardiography from optical sensing using lightweight neural network." IEEE Transactions on Artificial Intelligence 5.7 (2024): 3535-3550.

[3] Vo, Khuong, Mostafa El-Khamy, and Yoojin Choi. "PPG-to-ECG Signal Translation for Continuous Atrial Fibrillation Detection via Attention-based Deep State-Space Modeling." 2024 46th Annual International Conference of the IEEE Engineering in Medicine and Biology Society (EMBC). IEEE, 2024.

[4] Nambu, Yuta, Masahiro Kohjima, and Ryuji Yamamoto. "CardioFlow: Learning to Generate ECG from PPG with Rectified Flow." ICASSP 2025-2025 IEEE International Conference on Acoustics, Speech and Signal Processing (ICASSP). IEEE, 2025.

---

> ### Author Response · Authors · 2025-11-26
> **To Reviewer brYy (Part I)**
>
> Thank you very much for your thoughtful feedback and constructive comments. We truly appreciate the opportunity to clarify key aspects of our work. We kindly request that you reconsider the score in light of the clarifications below.
>
> **Weakness1/Question1: The work has weak justification on the given task when it remains physiologically infeasible to use PPG for guiding ECG generation process due to their different nature. Please discuss if PPG to ECG translation is theoretically reasonable.**
>
> We respectfully disagree that using PPG to guide ECG generation is "physiologically infeasible." PPG and ECG are not independent signals of unrelated nature, but two time-locked projections of the same cardiac cycle: ECG reflects the primary electrical activation of the myocardium, while PPG measures the resulting peripheral hemodynamic pulse. This electrical–mechanical coupling is precisely why PPG is routinely used for heart-rate and rhythm monitoring, and why recent studies have already demonstrated $PPG \to ECG$ translation and PPG-based ECG surrogates in remote-monitoring and low-resource settings [1–3]. In our setting, this relationship is further constrained at the data level: PPG and ECG are synchronously recorded on the same clinical monitoring system with a shared hardware clock, and each 10-second PPG segment is paired with a unique, time-registered Lead II ECG segment extracted from the identical time interval.
>
> Within PPGFlowECG, this physiological link is not only assumed but enforced by the learning objective. CardioAlign is trained with paired PPG–ECG supervision together with symmetric KL alignment, contrastive alignment, and cross-reconstruction losses, restricting the generator to model shared cardiovascular factors (e.g., heart rate, rhythm regularity, gross morphology) rather than arbitrary waveforms. The resulting ECG surrogates are evaluated quantitatively and in clinically motivated tasks. On MCMED, they achieve a Macro-AUROC of 0.631 for multi-label CVD detection (Table 2); on MIMIC-AFib, they support strong AF detection when used as input to a specialist ECG model (Table 8); heart-rate estimation from generated ECG outperforms that from raw PPG (Figure 8); and in the cardiologist Turing-style experiment, generated ECGs are judged comparable to real ECGs while slightly improving AF diagnostic performance when combined with PPG.
>
> Taken together with prior $PPG \to ECG$ studies [1–3], these results show that, under synchronized acquisition and explicit latent constraints, PPG contains sufficient cardiovascular information to reconstruct an ECG surrogate that is both quantitatively accurate and clinically useful. In this sense, using PPG to guide ECG generation in PPGFlowECG is physiologically grounded rather than infeasible.
>
> [1] P. Sarkar and A. Etemad. Cardiogan: Attentive generative adversarial network with dual discriminators for synthesis of ECG from PPG. AAAI, 2021.
>
> [2] E. Lan. Performer: A novel PPG-to-ECG reconstruction transformer for a digital biomarker of cardiovascular disease detection. WACV, 2023.
>
> [3] D. Shome, P. Sarkar, and A. Etemad. Region-disentangled diffusion model for high-fidelity PPG-to-ECG translation. AAAI, 2024.
>
> **Weakness2: The motivation of using clinical data for translation is questionable.**
>
> Our choice of clinical paired PPG–ECG data is driven by data availability and clinical validity rather than convenience. At present, large-scale, synchronously recorded PPG–ECG pairs with reliable cardiovascular diagnoses exist almost exclusively in clinical environments (ED, ICU, peri-operative, AF cohorts). In free-living or home settings, such paired recordings are extremely scarce, and when they do exist they typically lack adjudicated disease labels, which makes it impossible to learn and quantitatively evaluate a pathology-aware $PPG \to ECG$ mapping.
>
> At the same time, our intended deployment scenario is exactly home and wearable monitoring, where PPG is continuously available and subjects spend most of their time in resting or low-activity states. These resting segments are physiologically close to the controlled, resting clinical recordings we use for training. Using clinical data therefore allows us to (i) exploit high-quality PPG–ECG synchrony and rich diagnostic labels to learn how cardiovascular pathologies modulate the ECG given the underlying hemodynamics, and (ii) transfer this mapping to home PPG streams under similar resting conditions, which is the primary use case we target. In this sense, clinical data are not a mismatched choice but a necessary and realistic proxy to obtain a clinically grounded $PPG \to ECG$ translator under the current data landscape. We will clarify this motivation and the resting-state deployment assumption in the revised manuscript.

---

> ### Author Response · Authors · 2025-11-26
> **To Reviewer brYy (Part II)**
>
> **Weakness3: The framework design as “Align first, Generate later” are not fundamentally novel. Also, the loss functions and ODE process are also introduced in prior works.**
>
> We agree that latent alignment and flow-based generation have each been studied before, but our novelty lies in a *physiology-specific* instantiation and its empirically validated necessity. CardioAlign jointly enforces distribution-level symmetric KL, instance-wise contrastive alignment, and cross-reconstruction to place PPG and ECG on a shared, low-curvature latent manifold tailored to cardiovascular factors, and we are, to our knowledge, the first to couple this aligned latent space with a rectified-flow ODE for $PPG \to ECG$ translation. Controlled baselines using the same rectified-flow architecture either (i) directly in ECG signal space (Tables 1–2) or (ii) in a latent space without CardioAlign (“w/o CardioAlign Encoder” in Tables 3–4) systematically underperform our full model in both signal fidelity and downstream CVD/AF metrics. Thus, while the individual components (losses and ODE solver) are known, their specific, physiologically grounded coupling and the consistent gains over data-space and “latent-only” variants constitute the core methodological contribution of this work.
>
> **Weakness4: Experimental results are not sound.**
>
> We respectfully disagree that the experimental results are not sound. Our evaluation spans four heterogeneous clinical datasets (ED, ICU, peri-operative, AF ICU) comprising over 11M paired $10$-second PPG–ECG segments, and all methods are trained and compared under a unified preprocessing and training protocol. Across these cohorts, PPGFlowECG consistently outperforms or matches strong GAN, diffusion, and flow baselines on signal-level metrics (MAE/RMSE/FD/FID/MAE_HR, Table 1) and achieves Macro-AUROC $= 0.631$ on six CVD categories on MCMED (Table 2). Component-wise ablations (Tables 3–4) show systematic degradation whenever CardioAlign losses are removed, indicating that the gains are not due to incidental training effects.
>
> Beyond internal evaluation, external validation demonstrates zero-shot transfer from MCMED to MIMIC-AFib with AF-detection $F1 = 0.82$ and $AUROC = 0.83$ (Table 10). Cardiologist Turing-style tests and AF-diagnosis studies (Tables 5–6) further indicate that synthesized ECGs are rated as perceptually comparable to real ECGs and are at least as useful for AF diagnosis when combined with PPG. Taken together, the large-scale, multi-cohort design, agreement between signal-level and task-level metrics, and controlled ablations and reader studies jointly support the soundness of our experimental conclusions.
>
> **Question1: How to handle when the generated ECGs contain wrong patterns (that might be serious in clinical setting).**
>
> We explicitly position PPGFlowECG as an auxiliary representation for PPG-only scenarios, not as a replacement for clinical ECG acquisition. In any setting where a real $12$-lead or single-lead ECG is available, that recording should remain the sole basis for diagnostic decisions; synthesized ECGs are either ignored or used only as internal algorithmic features. In PPG-only settings, our use case is not to display the generated ECG as a standalone diagnostic trace, but to feed it into validated downstream detectors whose performance is quantified against adjudicated labels (e.g., AF and multi-label CVD). In deployment, this can be combined with conservative operating points and simple consistency checks (e.g., agreement between PPG-derived and ECG-derived heart rate) so that the system abstains or defers to clinical evaluation when the synthesized ECG behaves atypically. We will clarify these intended-use and safety constraints in the revised manuscript.

---

> ### Author Response · Authors · 2025-11-26
> **To Reviewer brYy (Part III)**
>
> **Question2: Related to the above point, a question raised is that do we really need the proposed translation framework with computationally cost complexity and latency (flow-based)? Please add an comparable experiment on using directly PPG encoder pretrained in Stage 1 (and/or other pretrained PPG encoder [1]) to finetune and evaluate on the downstream tasks, without relying on ECG or the generation process.**
>
> We respectfully disagree that a direct PPG encoder could replace our translation framework. First, the cardiologist study already contains an implicit "PPG-only" condition: reading PPG alone yields an F1 of $0.77$, whereas adding real ECG increases F1 to $0.93$, and substituting the real ECG with the generated ECG from PPGFlowECG further increases it to $0.94$. This indicates that the synthesized ECG exposes diagnostic structure that is not accessible from PPG alone, even to human experts.
>
> Following the reviewer’s suggestion, we additionally implemented a PPG-only baseline that uses the Stage-1 PPG encoder directly for downstream prediction, without ECG generation, under the same data splits and training protocol as PPGFlowECG. As shown below, this PPG-only model consistently underperforms PPGFlowECG across all six ICD categories and in Macro-AUROC:
>
> | Methods | Test Modality | I48 | I71 | I70 | I44 | I25 | I50 | Macro-AUROC |
> | :--- | :---: | :---: | :---: | :---: | :---: | :---: | :---: | :---: |
> | PPG Encoder pretrained in Stage 1 | PPG | 0.638 | 0.592 | 0.573 | 0.565 | 0.558 | 0.549 | 0.579 |
> | **PPGFlowECG** | Gen. ECG | **0.708** | **0.626** | **0.622** | **0.619** | **0.608** | **0.604** | **0.631** |
>
> **Question3: Add another experiment without stage 1 to compare with current results (With stage 2 only, and unfrozen the Encoder and Decoder.).**
>
> We thank the reviewer for the suggestion. The ablation "w/o CardioAlign Encoder" in Tables 3–4 already corresponds to this setting: Stage 1 is removed, and in Stage 2 we train separate PPG and ECG encoders together with the ECG decoder and rectified flow *from scratch*, with all these components unfrozen and optimized jointly, without any CardioAlign pretraining. This Stage-2–only variant consistently underperforms the full two-stage PPGFlowECG on both signal-level and diagnostic metrics, indicating that the proposed Stage-1 alignment brings clear benefits rather than being redundant. We will clarify in the revised manuscript that "w/o CardioAlign Encoder" exactly implements the requested Stage-2–only configuration.
>
> **Question4: Related to the above point, in the proposed framework, since the ECG decoder in stage 2 is frozen, what is exactly the input of it in this stage? Is it different from the input of the ECG decoder in stage 1 (from 2 sourses: PPG and ECG)?**
>
> In Stage 1, the ECG decoder directly takes encoder latents from both modalities for self-reconstruction ($z_{\text{ECG}} \to \hat{\text{ECG}}$) and cross-reconstruction ($z_{\text{PPG}} \to \tilde{\text{ECG}}$). In Stage 2, the CardioAlign Encoder is frozen and the decoder only receives flow-generated ECG latents $z_{\text{ECG}}$ produced by the conditional rectified flow conditioned on $z_{\text{PPG}}$, i.e., the PPG signal no longer enters the decoder directly.
>
> **Question5: Why does the work only focus on single-lead II generation and on clinical datasets? Is the motivation of using clinical data for translation questionable?**
>
> We respectfully disagree that our focus on single-lead II clinical data undermines the relevance of the framework. Lead II is the clinical reference channel for rhythm assessment, and its use is standard in prior $PPG \to ECG$ translation work. Extending CardioAlign + latent rectified flow to $12$-lead ECGs is architecturally straightforward (by adding lead-wise channels/embeddings), but at present there is no large-scale, labeled, paired PPG–$12$-lead resource comparable to MCMED on which to train and evaluate a pathology-aware model.
>
> Our use of clinical paired PPG–ECG data is driven by data availability and clinical validity rather than by an in-hospital deployment target. Large, synchronously recorded PPG–ECG pairs with adjudicated cardiovascular diagnoses exist almost exclusively in clinical settings. In free-living or home environments, paired PPG–ECG recordings are extremely scarce and, when available, are essentially unlabeled, which makes it impossible to learn or quantitatively assess a disease-aware $PPG \to ECG$ translator. We do not propose to replace clinical ECG when it is available; instead, our intended deployment scenario is PPG-only wearable/home monitoring, where subjects spend most of their time in resting or low-activity states that are physiologically close to the resting clinical recordings used for training. In this sense, clinical data serve as a necessary and realistic proxy to obtain a clinically grounded $PPG \to ECG$ mapping that can then be applied to home PPG streams in similar resting conditions.

---

> ### Author Response · Authors · 2025-11-26
> **To Reviewer brYy (Part IV)**
>
> **Question6: Line 459-461: adding more justification would be better why gen is better than real here? Also, why just use 50 signals (25 AF, 25 non-AF) in this experiment (which is too few for showing generalisation)?**
>
> In the current draft we wrote that “the generated signals convey diagnostic information equivalent to—and in this study even surpassing—real ECGs.” We agree that this wording is too strong. As clarified in the main text, our intended conclusion is *non-inferiority* rather than that synthetic ECGs are intrinsically “better” than real ones. The F1 score increases only from $0.93$ (PPG + real ECG) to $0.94$ (PPG + generated ECG), which is within typical inter-reader variability. We hypothesize that this small gain arises because the generated ECG acts as a denoised, morphology-regularized view of the same rhythm, consistent with its strong signal-level fidelity and AF-classification performance on MIMIC-AFib, but we will explicitly soften the statement and not present this difference as statistically or clinically decisive.
>
> Regarding the sample size, the AF reader study is designed as a focused, controlled Turing-test–style experiment rather than as the sole evidence of generalisation. Although the case pool contains $50$ signals ($25$ AF / $25$ non-AF), each case is evaluated independently by $5$ cardiologists under $3$ viewing conditions, yielding $750$ binary decisions in total, which is in line with typical scales of cardiology reader studies. Moreover, this experiment complements, rather than replaces, our large-scale quantitative evaluation on millions of clinical segments across four datasets (Tables 1–2, 8–10); in the revision, we will clarify that its role is to demonstrate that synthesized ECGs are perceptually realistic and at least as diagnostically useful as real ECGs under controlled conditions, not to establish population-level generalisation on its own.
>
> **Question7: Please discuss the differences and provide possible comparisons with [2,3,4].**
>
> Li et al. [2] and Vo et al. [3] both learn direct $PPG \to ECG$ mappings in the signal domain using task-specific architectures—a lightweight multi-scale CNN with diagnosis-oriented training in [2], and an attention-based latent state-space model for continuous AF detection in [3]. Unlike our work, neither constructs a shared latent physiological manifold between PPG and ECG, nor employs flow-based generative dynamics. CardioFlow [4] is closer in methodology through its use of rectified flow, but it operates directly in ECG signal space on relatively small emotion/exercise datasets and does not incorporate multi-level cross-modal alignment or large-scale multi-disease evaluation.
>
> Our work differs along two key axes: (i) CardioAlign, a physiologically grounded multi-level latent alignment module (symmetric KL, contrastive, cross-reconstruction) that forms a unified PPG–ECG latent manifold; and (ii) latent rectified flow, trained within this manifold. Our controlled baselines include a "direct data-domain flow" setting analogous to [4] as well as GAN and diffusion models, and consistently show substantial gaps in both signal-level and diagnostic metrics.
>
> We do not include [2–4] as numerical baselines because their full training code and configurations are not publicly available, making faithful re-implementation difficult. Instead, we compare against strong representatives of the same families (GAN/diffusion/flow) under a unified pipeline. Together with our multi-cohort, multi-label CVD evaluation, external AF validation, and cardiologist studies, this establishes a distinct problem setting and contribution relative to [2–4]. Our experiments already cover the axes requested by the reviewer: data- vs. latent-domain flow and no/single- vs. multi-level alignment.
>
> [2] Li, Yuenan, et al. "Inferring electrocardiography from optical sensing using lightweight neural network." IEEE Transactions on Artificial Intelligence 5.7 (2024): 3535-3550.
>
> [3] Vo, Khuong, Mostafa El-Khamy, and Yoojin Choi. "PPG-to-ECG Signal Translation for Continuous Atrial Fibrillation Detection via Attention-based Deep State-Space Modeling." 2024 46th Annual International Conference of the IEEE Engineering in Medicine and Biology Society (EMBC). IEEE, 2024.
>
> [4] Nambu, Yuta, Masahiro Kohjima, and Ryuji Yamamoto. "CardioFlow: Learning to Generate ECG from PPG with Rectified Flow." ICASSP 2025-2025 IEEE International Conference on Acoustics, Speech and Signal Processing (ICASSP). IEEE, 2025.

---

> ### Author Response · Authors · 2025-11-26
> **To Reviewer brYy (Part V)**
>
> **Question8: The seen improvements when testing ablation studies are so minor, showing weak contributions of different loss functions (e.g., Table 3, 4).**
>
> To directly assess whether each loss contributes meaningfully, we performed an additional ablation isolating the three alignment terms. As shown below, every loss provides a clear improvement over the “separate latent space generation’’ baseline, and the full combination attains the best performance:
> | Settings | MAE_HR | Macro-AUROC |
> | :--- | :---: | :---: |
> | Separate latent space generation | 2.12 | 0.589 |
> | w $\mathcal{L}_{\text{align}}$ | 1.84 | 0.624 |
> | w $\mathcal{L}_{\text{infoNCE}}$ | 1.89 | 0.617 |
> | w $\mathcal{L}_{\text{cross}}$ | 1.97 | 0.608 |
> | Complete | **1.80** | **0.631** |
>
> **Question9: Provide additional experiments on different modeling approaches (e.g., diffusion-based) in stage 2, besides ODE.**
>
> We appreciate the suggestion and have conducted an additional experiment where Stage 2 is replaced by a diffusion-based sampler. Concretely, we implement a latent diffusion model operating in the *same* CardioAlign latent space, with the ECG encoder/decoder and all training hyperparameters kept identical to the rectified-flow setting. The only change is the generative family in Stage 2. As shown in Table, latent diffusion underperforms latent rectified flow on both signal-level and diagnostic metrics, while requiring substantially more sampling steps:
>
> | Methods | MAE | RMSE | FD | FID | MAE_HR |
> | :--- | :---: | :---: | :---: | :---: | :---: |
> | Diffusion-based ($T=10$) | $0.82$ | $1.26$ | $48.12$ | $15.63$ | $1.87$ |
> | **Flow-based** ($T=10$) | $\mathbf{0.73}$ | $\mathbf{1.14}$ | $\mathbf{43.99}$ | $\mathbf{12.84}$ | $\mathbf{1.80}$ |
>
> | Methods | I48 | I71 | I70 | I44 | I25 | I50 | Macro-AUROC |
> | :--- | :---: | :---: | :---: | :---: | :---: | :---: | :---: |
> | Diffusion-based ($T=10$) | $0.681$ | $0.613$ | $0.601$ | $0.604$ | $0.592$ | $0.595$ | $0.614$ |
> | **Flow-based** ($T=10$) | $\mathbf{0.708}$ | $\mathbf{0.626}$ | $\mathbf{0.622}$ | $\mathbf{0.619}$ | $\mathbf{0.608}$ | $\mathbf{0.604}$ | $\mathbf{0.631}$ |
>
> **Question10: Section 4.5 seems to be overclaim. Figure 5 actually doesn’t show expected insights when the attention looks different and the signals also looks different between the original and generated ones.**
>
> We appreciate the concern and agree that the wording in Sec. 4.5 may sound stronger than what Figure 5 visually supports; we will soften the claim accordingly. Our intention was not to imply pixel-level waveform identity, but to show that, across multiple CVD categories, Grad-CAM on generated ECGs highlights the same diagnostically relevant regions as real ECGs, even when amplitude or fine morphology differ.
>
> To clarify this, we add a quantitative measure: the IoU of the top-20% Grad-CAM saliency between real and generated ECGs, which captures the overlap of high-attention temporal regions. As shown below, PPGFlowECG achieves the best or second-best IoU across all six ICD-10 categories, substantially outperforming GAN, diffusion, and flow baselines:
>
> | Methods | I48 | I71 | I70 | I44 | I25 | I50 |
> | :--- | :---: | :---: | :---: | :---: | :---: | :---: |
> | CardioGAN | 0.48 | **0.52** | 0.45 | 0.44 | 0.46 | 0.43 |
> | DDPM ($T=50$) | 0.52 | 0.47 | 0.46 | **0.49** | 0.47 | 0.49 |
> | RDDM ($T=10$) | 0.46 | 0.45 | 0.43 | 0.44 | 0.42 | 0.42 |
> | Rectified Flow ($T=10$) | 0.39 | 0.41 | 0.39 | 0.37 | 0.38 | 0.39 |
> | **PPGFlowECG** ($T=10$) | **0.61** | 0.50 | **0.54** | **0.49** | **0.50** | **0.51** |

---

### Official Review · Reviewer_UoNC · 2025-10-27

**Soundness:** 3
**Presentation:** 3
**Contribution:** 3
**Rating:** 6
**Confidence:** 4

**Summary:**

The paper proposes PPGFlowECG: a two-stage framework that first uses a shared CardioAlign encoder to align PPG/ECG to a unified latent space, and then performs conditional Rectified Flow on this latent space to generate ECG. Signal reconstruction and downstream CVD diagnostic evaluation are performed on multiple datasets such as MCMED, including clinical Turing Test and doctor interpretation experiments.

**Strengths:**

1. A clear "align first, generate later" design: shared encoder + distribution/instance alignment + cross-modal reconstruction, combined with rectified latent space flow, balances interpretability and sampling efficiency; outperforming GAN/diffusion/data domain flow baselines on multiple reconstruction metrics and downstream diagnostics.

2. Comprehensive evaluation: Four datasets (including clinical-grade MCMED, 10M+ aligned segments), unified preprocessing, providing MAE/RMSE/FD/FID/heart rate error and Macro-AUROC for multiple diseases, and performing ablation and external zero-shot validation.

3. Clinical usability signal: Turing Test is close to random (indistinguishable from real or fake), and replacing the real ECG with synthetic ECG can still improve/maintain the doctor's AF diagnosis F1 (0.94).

**Weaknesses:**

1. The boundaries of novelty need to be more clearly defined: Compared to the existing approach of "directly using Flow/diffusion in the data domain", this article's latent space Flow + multiple alignment is a combined innovation; it is recommended to make a more systematic comparison with "separate latent space generation or single alignment strategy" to strengthen the necessity argument.

2. Metric selection and consistency: In the external validation, signal-level metrics showed an "inconsistent metric criteria" phenomenon, with FD improving but FID/HR deteriorating. The authors' verbal explanation was reasonable, but they still recommended further analysis (such as correspondence with clinical factors).

3. Data protocol details: Except for MCMED, which uses the official split, other datasets use a subject-level 80/20 random split; it is recommended to disclose the number of subjects/window independence and repeated trial variance for each episode; currently only the main table with a single step size of T=10 is reported. The impact of the number of sampling steps is in the appendix, and a sentence can be added to the main text.

4. Clinical coverage: Only Lead II with a 10-s window is reported; generalization to rhythm-morphology complex lesions in multiple leads/longer time series remains unknown.

**Questions:**

1. Can the key tables be supplemented with multiple seed ± std/significance? Especially Table 1/Table 2 and the doctor reading experiment.

2. Deconstructing the sources of sliders for FD/FID/MAEHR in external validation (data domain differences, basic gradient distribution, noise spectrum)?

3. If I switch to multi-lead ECG or longer clips, what changes are needed in CardioAlign (does the shared encoder still hold true)?

4. If I switch to multi-lead ECG or longer clips, what changes are needed in CardioAlign (does the shared encoder still hold true)?

5. In this article, ECG is generated by PPG, which means it is essentially a process of information addition. Can you further explain why the additional information can improve the accuracy of the diagnostic results?

---

> ### Author Response · Authors · 2025-11-26
> **To Reviewer UoNC (Part I)**
>
> Thank you for your positive evaluation of our work. Below we provide a point-by-point response to your questions and hope it clarifies all of your concerns.
>
> **Weakness 1: It is recommended to make a more systematic comparison with "separate latent space generation or single alignment strategy" to strengthen the necessity argument.**
>
> We thank the reviewer for the helpful suggestion. For clarification, **the ablation "w/o CardioAlign Encoder" in Table 3 already represents a latent rectified-flow baseline**: the rectified flow module remains unchanged, but CardioAlign is removed and replaced with two independent encoders, resulting in no cross-modal latent alignment. In line with the reviewer’s recommendation, we further perform a systematic latent-space comparison by enabling only one alignment component at a time. The results are summarized below:
>
> | Settings | MAE_HR | Macro-AUROC |
> | :--- | :---: | :---: |
> | Separate latent space generation | 2.12 | 0.589 |
> | w $\mathcal{L}_{\text{align}}$ | 1.84 | 0.624 |
> | w $\mathcal{L}_{\text{infoNCE}}$ | 1.89 | 0.617 |
> | w $\mathcal{L}_{\text{cross}}$ | 1.97 | 0.608 |
> | Complete | **1.80** | **0.631** |
>
> **Weakness2: In the external validation, signal-level metrics showed an "inconsistent metric criteria" phenomenon, with FD improving but FID/HR deteriorating. The authors' verbal explanation was reasonable, but they still recommended further analysis (such as correspondence with clinical factors).**
>
> We thank the reviewer for the helpful suggestion. As shown in Table 10, downstream AF detection remains strong under external validation (F1 0.82, AUROC 0.83 vs. 0.87/0.87 for the specialist model), indicating that clinically relevant pathological features are preserved despite shifts in low-level waveform statistics. In our case, FD is more sensitive to global morphology, while FID/HR are strongly affected by differences in noise spectrum, baseline drift and heart-rate distribution between MCMED and MIMIC; this explains why FD improves while FID/HR partially deteriorate. In the revision, we will explicitly state that clinical performance is the primary consistency criterion.

---

> ### Author Response · Authors · 2025-11-26
> **To Reviewer UoNC (Part II)**
>
> **Weakness3/Question1: It is recommended to disclose the number of subjects/window independence and repeated trial variance for each episode; currently only the main table with a single step size of T=10 is reported. The impact of the number of sampling steps is in the appendix, and a sentence can be added to the main text.**
>
> We thank the reviewer for the helpful suggestions. All continuous signals are segmented into non-overlapping $10$-second windows, and train/test splits are performed at the subject level, ensuring that no subject contributes windows to multiple splits. The subject-level statistics for all datasets (except MCMED, which uses the official split) are reported in the table below.
>
> | Datasets | Train Subjects | Train Windows | Test Subjects | Test Windows |
> | :--- | :---: | :---: | :---: | :---: |
> | VitalDB | 5110 | 1007165 | 1278 | 10720 |
> | MIMIC-AFib | 28 | 2044 | 7 | 577 |
> | BIDMC | 42 | 1458 | 11 | 326 |
>
> To address the reviewer’s concerns regarding repeated-trial variance and statistical significance, we additionally report mean $\pm$ std over five random seeds for the key metrics on the MCMED dataset (Tables 1 and 2). These results are shown in the table below.
>
> | Methods | MAE | RMSE | FD | FID | $MAE\_HR$ |
> | :--- | :---: | :---: | :---: | :---: | :---: |
> | CardioGAN | $0.97\pm0.02$ | $1.41\pm0.05$ | $80.72\pm2.53$ | $53.17\pm1.83$ | $2.76\pm0.12$ |
> | DDPM ($T=50$) | $0.95\pm0.03$ | $1.35\pm0.04$ | $\mathbf{13.15\pm1.27}$ | $45.01\pm2.73$ | $7.03\pm0.75$ |
> | RDDM ($T=10$) | $0.99\pm0.02$ | $1.40\pm0.04$ | $55.84\pm4.19$ | $21.32\pm2.18$ | $7.51\pm0.38$ |
> | Rectified Flow ($T=10$) | $0.98\pm0.01$ | $1.36\pm0.02$ | $105.49\pm3.85$ | $53.76\pm3.05$ | $24.60\pm2.20$ |
> | **PPGFlowECG** ($T=10$) | $\mathbf{0.73\pm0.02}$ | $\mathbf{1.15\pm0.03}$ | $42.67\pm3.06$ | $\mathbf{11.94\pm1.67}$ | $\mathbf{1.83\pm0.16}$ |
>
> | Methods | I48 | I71 | I70 | I44 | I25 | I50 | Macro-AUROC |
> | :--- | :---: | :---: | :---: | :---: | :---: | :---: | :---: |
> | CardioGAN | $0.608\pm0.015$ | $\mathbf{0.654\pm0.018}$ | $0.568\pm0.022$ | $0.570\pm0.011$ | $0.573\pm0.013$ | $0.537\pm0.022$ | $0.585\pm0.015$ |
> | DDPM ($T=50$) | $0.664\pm0.018$ | $0.601\pm0.015$ | $0.578\pm0.018$ | $\mathbf{0.626\pm0.014}$ | $0.581\pm0.011$ | $0.593\pm0.017$ | $0.607\pm0.014$ |
> | RDDM ($T=10$) | $0.576\pm0.020$ | $0.578\pm0.012$ | $0.539\pm0.022$ | $0.573\pm0.019$ | $0.547\pm0.015$ | $0.523\pm0.020$ | $0.556\pm0.016$ |
> | Rectified Flow ($T=10$) | $0.501\pm0.014$ | $0.505\pm0.018$ | $0.504\pm0.010$ | $0.498\pm0.013$ | $0.498\pm0.022$ | $0.499\pm0.014$ | $0.501\pm0.014$ |
> | **PPGFlowECG** ($T=10$) | $\mathbf{0.706\pm0.012}$ | $0.628\pm0.011$ | $\mathbf{0.624\pm0.018}$ | $0.617\pm0.016$ | $\mathbf{0.609\pm0.015}$ | $\mathbf{0.605\pm0.017}$ | $\mathbf{0.632\pm0.013}$ |
>
> For the Clinical Turing Test (Table 5), we report accuracy, 95% exact binomial confidence intervals, and statistical significance vs. random guessing ($0.50$). For 4/5 cardiologists, accuracy is not statistically different from chance ($p>0.05$), indicating that synthetic ECGs are indistinguishable from real ECGs. Table 6 further reports the mean $\pm$ std. The corresponding summaries are provided in the tables below.
>
> | Cardiologist | Accuracy | 95% CI | p-value (vs 0.5) |
> | :---: | :---: | :---: | :---: |
> | 1 | 0.44 | $[0.300, 0.587]$ | 0.4799 |
> | 2 | 0.70 | $[0.554, 0.821]$ | 0.0066 |
> | 3 | 0.50 | $[0.355, 0.645]$ | 1.0000 |
> | 4 | 0.50 | $[0.355, 0.645]$ | 1.0000 |
> | 5 | 0.44 | $[0.300, 0.587]$ | 0.4799 |
>
> | Test Modality | Accuracy | Sensitivity | Specificity | F1 Score |
> | :--- | :---: | :---: | :---: | :---: |
> | PPG only | $0.84\pm0.15$ | $0.70\pm0.32$ | $\mathbf{0.98\pm0.03}$ | $0.77\pm0.28$ |
> | PPG + Real ECG | $\mathbf{0.93\pm0.08}$ | $0.90\pm0.12$ | $0.97\pm0.05$ | $0.93\pm0.09$ |
> | PPG + Gen. ECG by **PPGFlowECG** | $\mathbf{0.93\pm0.07}$ | $\mathbf{0.96\pm0.04}$ | $0.90\pm0.15$ | $\mathbf{0.94\pm0.05}$ |
>
> Finally, regarding the sampling protocol, Appendix D.5 already includes an ablation over the number of ODE steps $T\in\{5,10,15,20,25\}$. We will add a sentence in the main text explicitly referring to this analysis.

---

> ### Author Response · Authors · 2025-11-26
> **To Reviewer UoNC (Part III)**
>
> **Weakness4/Question3/4: Only Lead II with a 10-s window is reported; generalization to rhythm-morphology complex lesions in multiple leads/longer time series remains unknown. If I switch to multi-lead ECG or longer clips, what changes are needed in CardioAlign (does the shared encoder still hold true)?**
>
> We appreciate the reviewer’s concern about generalizing to more complex rhythms and longer ECG segments. We agree that our current experiments focus on single-lead (Lead II) 10-second segments, but this choice is deliberate and practical for several reasons. Lead II is the clinical reference channel for rhythm assessment, and its use is standard in prior PPG$\rightarrow$ECG translation work. The 10-second duration follows the conventional resting ECG protocol used in clinical reporting and guidelines, providing several consecutive cardiac cycles that are sufficient for rhythm assessment and basic morphology inspection, while remaining comparable across patients and cohorts. Additionally, it is used consistently across four heterogeneous clinical cohorts (ED, ICU, peri-operative, and AF ICU), which already span diverse clinical settings. Importantly, while our current evaluation uses Lead II, it is not limited to simple rhythms: MCMED includes six cardiovascular disease categories (rhythm, conduction, vascular, and structural) that require the model to capture subtle morphological abnormalities.
>
> Extending the CardioAlign framework to multi-lead or longer clips would mainly require architectural adjustments to the ECG encoder (more input channels, longer sequences, and optional lead/time embeddings), while the shared-encoder principle and CardioAlign loss remain unchanged. Our method is therefore compatible with multi-lead and long-clip settings by construction. The present work focuses on single-lead 10-s windows primarily because large-scale paired PPG–multi-lead datasets comparable to MCMED are not yet available. We will clarify this extensibility and the rationale for the 10-second design choice in the revised manuscript.
>
> **Question2: Deconstructing the sources of sliders for FD/FID/MAE_HR in external validation (data domain differences, basic gradient distribution, noise spectrum)?**
>
> We agree that the external-validation behavior of FD/FID/MAE_HR is influenced by multiple factors. As discussed in Appendix D.2, these metrics probe different aspects of the signal distribution and therefore do not form a single monotonic criterion. FD reflects low-order amplitude statistics (e.g., baseline level, variance, noise floor) and is sensitive to calibration and drift, whereas FID operates in an ECG-embedding feature space that emphasizes rhythm and morphology and is more affected by domain shift and mode collapse. MAE_HR is additionally driven by differences in heart-rate range and label distribution across cohorts. As summarized in Table 7, cross-dataset variations in sampling rate, amplitude scaling, and acquisition setting produce changes in baseline, motion artefacts, and rhythm distributions, which perturb these metrics in different and sometimes opposite directions.
>
> A full spectral or feature-space decomposition of FD/FID would indeed be informative but is beyond the scope of this work. In the revision, we will make explicit that (i) FD, FID, and MAE_HR arise from distinct feature spaces with different sensitivities to domain statistics, and (ii) clinical performance (e.g., AF detection, multi-label CVD classification) is the primary measure of consistency under external validation, with FD/FID/MAE_HR serving as complementary but domain-sensitive indicators.
>
> **Question5: In this article, ECG is generated by PPG, which means it is essentially a process of information addition. Can you further explain why the additional information can improve the accuracy of the diagnostic results?**
>
> We thank the reviewer for raising this important point. Our claim is not that PPGFlowECG accesses physiological information beyond what is present in the PPG, but that it reparameterizes the same underlying cardiac dynamics into an ECG-like representation that is better aligned with clinical decision rules.
>
> During training, CardioAlign uses paired PPG–ECG supervision together with symmetric KL, contrastive alignment, and cross-reconstruction to ensure that its latent code retains shared physiological factors (e.g., heart rate, rhythm structure, beat morphology) while suppressing modality-specific artefacts. At test time, conditioning on PPG alone, the rectified flow projects the input onto this ECG manifold, functioning as (i) a denoiser of PPG-specific noise (motion, sensor drift) and (ii) a task-aligned feature space in which ECG-like patterns are more easily separated by diagnostic classifiers. This mechanism explains the performance gains without invoking any additional physiological information.

---

### Official Review · Reviewer_5jd3 · 2025-10-28

**Soundness:** 2
**Presentation:** 3
**Contribution:** 2
**Rating:** 2
**Confidence:** 4

**Summary:**

PPGFlowECG is a two-stage translation framework: it first aligns PPG and ECG in a shared latent space via CardioAlign, then uses a rectified flow in latent space that deterministically transports PPG latents to ECG latents to reconstruct waveforms.
The model demonstrates effectiveness on multiple public, clinically oriented datasets using signal-quality metrics and downstream diagnostic tasks.
Clinician evaluations and real diagnostic tasks indicate that generated ECGs provide useful auxiliary value, suggesting utility approaching that of real ECGs.

**Strengths:**

- The architecture and training procedure are staged and clear, making the paper easy to follow.

- Data composition, preprocessing, and evaluation protocols are systematically organized, and implementation details are sufficient for high reproducibility.

- On downstream tasks, the method shows competitive performance for multi-label cardiovascular disease classification and AF detection, and ablation studies disentangle the contributions of each loss and the sampling configuration.

- Clinician assessments confirm perceptual fidelity and clinical support value of the generated ECGs, and Grad-CAM offers qualitative evidence that diagnostic attention regions are similar between real and generated ECGs.

**Weaknesses:**

- The core design couples cross-modal PPG to ECG translation with shared latent alignment and latent rectified flow, reading as a refined integration of existing alignment/generation frameworks. The authors position this as an align-first, generate-later contribution.

- Grad-CAM visualizations show alignment of diagnostic attention, but no dedicated quantitative interpretability metrics are reported.

- Explanations for why the latent flow improves physiological mapping rely more on intuition and empirical results than formal analysis.

- Comparisons against GAN/diffusion/flow baselines and extensive ablations are solid, but adding signal-level registration (e.g., DTW), VAE-based models, and linear latent alignment baselines would strengthen the case.

**Questions:**

It would be helpful if the authors could address this weakness, if possible.

---

> ### Author Response · Authors · 2025-11-25
> **To Reviewer 5jd3 (Part I)**
>
> Thank you very much for your thoughtful feedback and constructive comments. We truly appreciate the opportunity to clarify key aspects of our work. We kindly request that you reconsider the score in light of the clarifications below.
>
> **Weakness1: The core design couples cross-modal PPG to ECG translation with shared latent alignment and latent rectified flow, reading as a refined integration of existing alignment/generation frameworks.**
>
> We respectfully disagree that PPGFlowECG is merely a refined combination of existing modules. Our contribution is concrete and arises from two physiologically motivated components that are jointly necessary for the result.
>
> First, **CardioAlign** is not a generic shared encoder: it performs (i) distribution-level alignment via symmetric KL and mean matching, (ii) instance-wise contrastive alignment, and (iii) cross-modal reconstruction, which embed PPG and ECG into a unified physiological latent manifold rather than loosely related feature spaces. Ablations (Table 3) show that removing CardioAlign or retaining only a single alignment term produces consistent drops in signal fidelity and multi-label CVD Macro-AUROC, demonstrating that the full multi-level alignment is essential.
>
> Second, we are the first to couple **latent rectified flow** with this aligned latent space for $PPG \to ECG$ translation. A controlled baseline using the same rectified-flow architecture directly in ECG data space (Table 2) fails to learn clinically meaningful mappings ($Macro\text{-}AUROC \approx 0.5$), whereas placing the identical flow in the CardioAlign latent manifold yields substantial improvements.
>
> Together, these results show that the novelty does not lie in the "align-first, generate-later" motif but in the specific, physiology-grounded instantiation and the empirically necessary coupling of multi-level CardioAlign with latent rectified flow—without which the observed physiological and clinical gains do not emerge.
>
> **Weakness2: Grad-CAM visualizations show alignment of diagnostic attention, but no dedicated quantitative interpretability metrics are reported.**
>
> In response to the reviewer’s comment, we additionally report a simple quantitative saliency measure: the IoU between the top-20% Grad-CAM regions of real and generated ECGs along the time axis. As shown in the table below, **PPGFlowECG** achieves the best or second-best real–gen IoU across all six ICD-10 categories, indicating that diagnostic attention is not only qualitatively consistent but also quantitatively well preserved.
>
> | Methods | I48 | I71 | I70 | I44 | I25 | I50 |
> | :--- | :---: | :---: | :---: | :---: | :---: | :---: |
> | CardioGAN | 0.48 | **0.52** | 0.45 | 0.44 | 0.46 | 0.43 |
> | DDPM ($T=50$) | 0.52 | 0.47 | 0.46 | **0.49** | 0.47 | 0.49 |
> | RDDM ($T=10$) | 0.46 | 0.45 | 0.43 | 0.44 | 0.42 | 0.42 |
> | Rectified Flow ($T=10$) | 0.39 | 0.41 | 0.39 | 0.37 | 0.38 | 0.39 |
> | **PPGFlowECG** ($T=10$) | **0.61** | 0.50 | **0.54** | **0.49** | **0.50** | **0.51** |
>
> **Weakness3: Explanations for why the latent flow improves physiological mapping rely more on intuition and empirical results than formal analysis.**
>
> We respectfully disagree that our explanation for the benefits of latent flow is purely intuitive. In Sec. 3.3 we formally define Stage 2 as learning a conditional rectified flow $v_{\theta}(x_t, t \mid z_{\mathrm{PPG}})$ in the CardioAlign latent space by minimizing
>
> $$\mathbb{E}\big\| v_{\theta}(x_t, t \mid z_{\mathrm{PPG}}) - (y - z) \big\|_2^2,$$
>
> where $x_t = (1-t)z + ty$ interpolates between a Gaussian prior $z \sim \mathcal{N}(0, I)$ and the ECG posterior $y$ conditioned on the same PPG segment. By construction, CardioAlign (Sec. 3.2) constrains PPG and ECG posteriors onto a shared, relatively low-curvature physiological manifold via symmetric KL, contrastive alignment, and cross-reconstruction. On such a manifold, the rectified-flow vector field is known to approximate the straightest displacement between the prior and posterior distributions, yielding a near-deterministic, geometry-aware transport map rather than an unconstrained stochastic sampler.
>
> The observed improvements are therefore not ad hoc, but consistent with this formal characterization: when the same rectified-flow architecture is trained directly in data space, the transport must operate on a much more irregular signal manifold and empirically suffers from higher trajectory curvature and mode confusion; in contrast, training in the aligned latent space learned by CardioAlign leads to better-behaved flows and, as shown by our controlled baselines and ablations (Tables 2--4), systematically superior signal fidelity and downstream CVD/AF performance compared with data-space RF or "latent without CardioAlign" variants. We will clarify this theoretical connection in the revised manuscript.

---

> ### Author Response · Authors · 2025-11-25
> **To Reviewer 5jd3 (Part II)**
>
> **Weakness4: Adding signal-level registration (e.g., DTW), VAE-based models, and linear latent alignment baselines would strengthen the case.**
>
> We appreciate the suggestion. However, DTW-based registration is not feasible in our setting: DTW has quadratic complexity in sequence length, and each $10$-second ($128$ Hz) segment must be compared across tens of thousands of pairs (e.g., $37,052$ test pairs on MCMED), leading to prohibitive time and memory requirements.
>
> In addition, we have implemented the reviewer’s other two requests: (i) a VAE-based $PPG \to ECG$ model, and (ii) a linear latent alignment baseline. The results on MCMED dataset are summarized below:
>
> | Methods | MAE | RMSE | FD | FID | MAE_HR |
> | :--- | :---: | :---: | :---: | :---: | :---: |
> | VAE | 0.90 | 1.33 | 70.35 | 32.63 | 5.50 |
> | Linear Latent Alignment | 0.85 | 1.31 | 58.56 | 23.47 | 3.43 |

---

### Note · Authors · 2026-01-20

I have read and agree with the venue's withdrawal policy on behalf of myself and my co-authors.